# Synthesizing Privacy-Preserving Text Data via Finetuning *without* Finetuning Billion-Scale LLMs

**Bowen Tan** [* 1]   **Zheng Xu** [2]   **Eric Xing** [1 3]   **Zhiting Hu** [4]   **Shanshan Wu** [2]

## Abstract

Synthetic data offers a promising path to train models while preserving data privacy. Differentially private (DP) finetuning of large language models (LLMs) as data generator is effective, but is impractical when computation resources are limited. Meanwhile, prompt-based methods such as private evolution (Xie et al., 2024; Hou et al., 2024) depend heavily on the manual prompts, and ineffectively use private information in their iterative data selection process. To overcome these limitations, we propose CTCL (Data Synthesis with **C**on**T**rollability and **CL**ustering), a novel framework for generating privacy-preserving synthetic data without extensive prompt engineering or billion-scale LLM finetuning. CTCL pretrains a lightweight 140M conditional generator and a clustering-based topic model on large-scale public data. To further adapt to the private domain, the generator is DP finetuned on private data for fine-grained textual information, while the topic model extracts a DP histogram representing distributional information. The DP generator then samples according to the DP histogram to synthesize a desired number of data examples. Evaluation across five diverse domains demonstrates the effectiveness of our framework, particularly in the strong privacy regime. Systematic ablation validates the design of each framework component and highlights the scalability of our approach.

## 1. Introduction

Many artificial intelligence (AI) applications improves their model performances by leveraging user data. For exam-

ple, models are improved by adapting to the typing text in user's mobile virtual keyboard (Hard et al., 2018; Xu et al., 2023), and aligning with user preference in a chatbot (OpenAI, 2024; Google, 2024; Llama Team, 2024). However, training models on user data raises privacy concerns, particularly in domains involving highly sensitive information, such as healthcare records (Milmo & Stacey, 2025) and chat messages (Silva, 2025). Researchers have shown that the training data can be memorized and potentially extracted from models (Carlini et al., 2021; Nasr et al., 2023; Carlini et al., 2023). Synthesizing privacy-preserving user data has emerged as a promising approach to mitigating these privacy risks. A popular approach is to differentially-private (DP) finetune a generative language model (LM) on user data, followed by generating synthetic data using the fine-tuned model (Bommasani et al., 2019; Putta et al., 2022; Mattern et al., 2022a; Yue et al., 2023). Benefiting from the development of open-sourced billion-scale large language models (LLMs) such as Llama (Touvron et al., 2023), DP-finetuned generators have demonstrated effectiveness in the downstream classification (Kurakin et al., 2023) and instruction tuning tasks (Yu et al., 2024). However, DP finetuning is both computationally expensive and resource-intensive, because it requires per-sample gradient operations in every training batch and large batch size to get good privacy-utility trade-off (Ponomareva et al., 2023). This results in higher memory usage and slower training speeds compared to the non-DP finetuning. Moreover, when the user data are decentralized across their own devices, and no centralized data collection is allowed following the data minimization privacy principle (McMahan et al., 2017; Kairouz et al., 2021; Daly et al., 2024), the devices performing local computations typically lack the necessary resources to finetune billion-scale LLMs.

To address the resource limitations, recent work has explored generating synthetic data that only require LLM API access, exemplified by the Private Evolution (PE) framework (Lin et al., 2024; Xie et al., 2024; Hou et al., 2024). These methods use an iterative process where samples are drawn from the LLMs using human-crafted prompts, and then filtered based on their similarity to the private data. This line of work has several limitations. First, they require prompt writers to have deep domain knowledge of the pri-

---

[*]Work done during an internship at Google.

[1]Carnegie Mellon University [2]Google Research [3]Mohamed bin Zayed University of Artificial Intelligence [4]UC San Diego. Correspondence to: Bowen Tan, Shanshan Wu <btan2@andrew.cmu.edu, shanshanw@google.com>.

*Proceedings of the 42nd International Conference on Machine Learning*, Vancouver, Canada. PMLR 267, 2025. Copyright 2025 by the author(s).

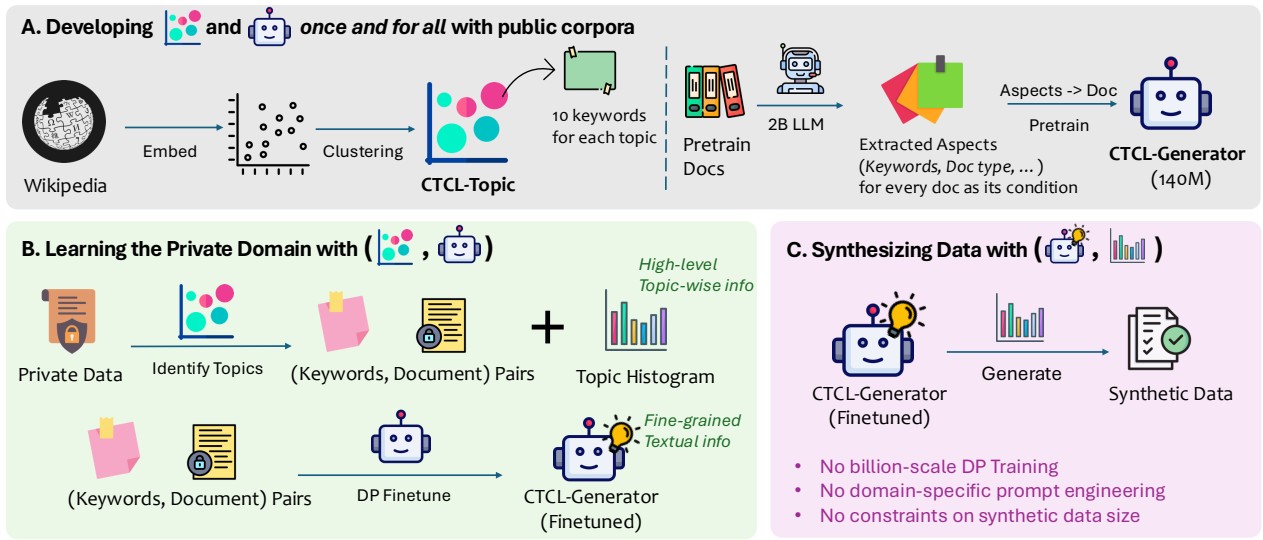

*Figure 1.* Overview of our CTCL (Data Synthesis with **ConT**rollability and **CL**ustering) framework: (**A**) A universal topic model and a lightweight 140M generator with strong controllability are developed *once and for all* on large-scale public corpora (§3.1 and §3.2); (**B**) To learn the private domain, we collect a DP topic histogram, and DP finetune the generator on the private data (§3.3); (**C**) Privacy-preserving synthetic data is generated based on the topic histogram and the finetuned generator (§3.4).

vate data, a requirement that can be unrealistic across diverse scenarios. They also heavily rely on the LLM's creativity and extensive prompt engineering tailored to the specific LLMs. More critically, the PE framework only uses the private data in the embedding space for example selection and filtration, and fails to fully leverage the fine-grained word-level information. This inherently limits the performance of the synthetic data in the downstream tasks, particularly the challenging generative tasks. As we will show in our experiments, unlike the standard classification tasks, these generative tasks are evaluated by next-word prediction accuracy, and hence, demand a finer-grained approximation of the private data distributions.

In this work, we introduce CTCL (Data Synthesis with **ConT**rollability and **CL**ustering), a novel framework for generating synthetic privacy-preserving data without finetuning billion-scale LLMs or domain-specific prompt engineering. As illustrated in Figure 1, CTCL comprises two key components: a lightweight 140M parameter generator and a universal topic model. Both components are pre-trained on the large-scale public corpora, SlimPajama (Soboleva et al., 2023) and Wikipedia (Foundation, 2023), respectively. When adapting to the private domains, the topic model produces a DP topic histogram to capture high-level distributional information, while the generator is DP finetuned to learn fine-grained, textual information. During the data generation phase, the DP-finetuned generator is sampled proportionally for each topic according to the DP topic histogram. An arbitrary amount of synthetic data can be generated by our CTCL-generator without paying additional

privacy costs, because of the post-processing property of DP (Dwork et al., 2014).

We validate our framework across five diverse downstream domains, including the medical contexts and human-to-human conversations, covering both generative tasks evaluated by the next-word prediction accuracy, and the standard classification tasks. Our framework demonstrates significant advantages over previous approaches, particularly under strict privacy constraints. Through a comprehensive analysis, we highlight the importance of each component in our design, and demonstrate the scalability of CTCL compared to prompting-based methods such as PE (Xie et al., 2024). *

## 2. Related Work

**Differential Privacy (DP)** Our operations on the private data adhere to the standard $(\epsilon, \delta)$-DP guarantee (Dwork et al., 2006), ensuring that the inclusion or exclusion of a single record has minimal impact on the algorithm's output. This constraint limits the model to learning generalizable patterns rather than memorizing individual data points. Specifically, we employ DP-Adam (Li et al., 2022; Yu et al., 2022) for DP finetuning, which clips per-sample gradients and injects Gaussian noise into each gradient update during training. We also add Gaussian noise to every bin when collecting DP histogram. For more details about the DP mechanism and the DP paremeters used in our experiments, see Appendix A and E.

---

| Prompt for OpenReview on GPT-3.5 (Xie et al., 2024) | Prompt for GBoard Dialogues on PaLM (Wu et al., 2024a) |
|---|---|
| Given the area and final decision of a research paper, you are required to provide a **detailed and long** review consisting of the following content:
1. briefly summarizing the paper in 3-5 sentences;
2. listing the strengths and weaknesses of the paper in details;
3. briefly summarizing the review in 3-5 sentences. | Imagine you are a female at age 23. You are using the Android Messages APP to message your family on your mobile phone on the afternoon of a vacation day. You want to chat about the following topic: I can't wait to come home and tell you all about it. Generate the conversation between you and your message receiver. |
| Prompt used in our pretraining data construction on Gemma-2-2B (§3.1) | |
| Describe this document in multiple aspects. Make sure "Document Type" and "Keywords" are two of the aspects. {document} | |

*Table 1.* The prompts used in existing synthetic data approaches versus in our pretraining data construction. Prompts in existing work usually requires in-depth domain knowledge and intensive prompt engineering specific to dataset and the LLM being prompted, while the one used in our data construction is simple and generally applicable on whatever types of documents in pretraining corpus.

**Synthetic Data via DP Finetuning of LMs**  This line of work DP finetunes an LM on the private data, and the finetuned LM is then used to generate synthetic data (Bommasani et al., 2019; Putta et al., 2022; Mattern et al., 2022a; Yue et al., 2023; Kurakin et al., 2023; Yu et al., 2024; Wang et al., 2024; Ochs & Habernal, 2024; Carranza et al., 2024). To preserve model capability under the DP training noise, these approaches often rely on billion-scale models, particularly for generative tasks. For instance, Yu et al. (2024) finetune LLaMA-7B (Touvron et al., 2023) with DP to generate short (usually single-sentence) human-to-machine instructions. In contrast, our framework incorporates a carefully designed learning process on the private data while using a significantly smaller backbone LM with only 140M parameters in DP finetuning. This substantially reduces the computational costs, making the approach more feasible for real-world resource-constrained applications.

**Synthetic Data via LLM API Prompting**  This line of research explores data synthesis using only LLM inference APIs, typically leveraging prompt engineering with domain-specific knowledge, such as specifying document structures or assuming roles (Wu et al., 2024a; Zhang et al., 2025). The Private Evolution (PE) framework (Lin et al., 2024; Xie et al., 2024; Hou et al., 2024) integrates the private information into the synthetic data through an iterative sample selection process. Specifically, the API-generated outputs are selected based on their proximity to private data measured by the differentially private nearest neighbors (DP-NN). In this setup, DP-NN serves as the sole mechanism for extracting information from the private data, limiting the extent to which its information is fully captured. Furthermore, the synthetic data size (which determines the number of bins in the DP-NN histogram) is often constrained in order to better tolerate the DP noise (see discussion in Appendix B). For instance, the synthetic datasets in (Xie et al., 2024) contain typically 2,000 to 5,000 examples across the experiments. Unlike the prompting-based methods, our framework does not require prompt engineering and prior domain knowledge when applied to downstream data. Additionally, our synthetic dataset size is not constrained, offering signifi-

cantly greater scalability compared to the PE approach. We discuss more related work including private inference in Appendix B.

## 3. CTCL Framework

In this work, we propose CTCL (Data Synthesis with **C**on**T**rollability and **CL**ustering), a framework for generating synthetic private data without requiring billion-scale DP finetuning or domain-specific prompt engineering. Figure 1 and Appendix H give an overview of CTCL.

Our framework consists of two key components: CTCL-Generator and CTCL-Topic. Both are developed only *once* using the large-scale public corpora. CTCL-Generator is a lightweight 140M-parameter conditional generator that supports free-form text input, allowing users to specify attributes such as keywords and document type (§3.1). The second component, CTCL-Topic, is a topic model that categorizes a given document into a predefined topic, represented by ten keywords (§3.2). To use these two components for learning a specific private domain: the topic model constructs a topic-wise histogram to capture high-level distributional information, while the generator is DP finetuned on private training data to retain low-level textual details (§3.3). After that, we use the DP finetuned generator and the DP topic histogram to produce an arbitrary number of synthetic samples without additional privacy costs (§3.4).

The design of our framework offers several advantages. First, compared with the existing billion-scale LLM DP finetuning, our backbone LM contains only 140M parameters, making DP finetuning practical for real-world resource-constrained applications. Second, unlike the prompting-based approaches that depend on hand-crafted domain-specific prompts that require in-depth expertise, our framework is applicable to any private domain regardless of prior domain knowledge. Third, PE-based methods need to balance between data quality and synthetic data size (see discussions in Appendix B), while our framework naturally allows for unlimited data samples using the DP finetuned generator,

| Document |
| --- |
| MORGANTOWN, W.Va. (November 11, 2015) – West Virginia University golf coach Sean Covich announced Wednesday that Ty Olinger (Blacksburg, Va.,/North Cross HS) and Etienne Papineau (St-Jean sur Richelieu, Quebec St-Lawrence) have committed to joining the Mountaineers starting in the fall of 2016. [...] |

| Extracted Aspects by Gemma-2-2B |
| --- |
| Tone : Informative, positive, celebratory, and official. |
| Style : Simple, straightforward, and direct. |
| Keywords : West Virginia University, Golf, Recruiting, College |
| Purpose : To announce a new recruiting class for WVU golf. |
| Structure : Follows a standard journalistic format |
| Document Type : Article, Sports News [...] |

*Table 2.* Example of the generated document description. It is used to form the `condition` part in our pretraining (`condition`, `document`) data corpus. This task of extracting existing information from the document doesn't require strong creativity, and hence, can be done by a relatively small LLM. The aspects marked in blue are *automatically* generated instead of pre-defined.

without additional privacy costs during generation.

The remainder of this section provides a detailed explanation of CTCL, covering its components (§3.1, §3.2), and the private learning and data synthesis processes (§3.3, §3.4).

### 3.1. CTCL-Generator

In our framework, CTCL-Generator is a lightweight (140M-parameter) conditional LM designed for strong controllability. Specifically, it accepts one or more feature assignments as input, and generates documents that adhere to these specifications. The assignments can include free-text inputs, such as *"Document Type: daily dialogue."* To enable this functionality within a small LM, we construct a large-scale dataset and perform continual pretraining of an unsupervisedly pretrained LM.

**Pretraining Data Curation**   We introduce a simple yet effective approach for constructing a large-scale condition-to-document corpus. Our method builds on SlimPajama (Soboleva et al., 2023), a large unsupervised pretraining corpus, and leverages a relatively small LLM, Gemma-2-2B (Team et al., 2024). Specifically, we employ a domain-agnostic and LLM-independent prompting strategy for each document in SlimPajama: "Describe the document in multiple aspects." This document description task is straightforward and not requiring the LLM's creativity, making it well-suited for a small LLM like Gemma-2-2B to efficiently handle the data construction process. As a result, we generate a large-scale pretraining dataset comprising 430M (`condition`, `document`) pairs, where the Gemma-2-2B generated document description is used as the `condition` part.

Tables 1 and 2 present the exact prompt we use and an

example of the generated document description. Notably, our prompt encourages the inclusion of "Document Type" and "Keywords" as aspects in the prompting output. This is designed to match how we use topic keywords to obtain high-level topic distributions, when adapting to a specific private domain (§3.3 and §3.4). Additionally, the document type is encouraged because it is the simplest high-level information to extract from the private data domain.

**Pretraining Setup**   We perform continual pretraining on top of BART-base (Lewis, 2019), a 140M-parameter sequence-to-sequence LM previously pretrained in an unsupervised manner. The model's encoder-decoder architecture is well suitable for conditioning on inputs through the encoder while generating outputs via the decoder. Optimization was performed using the AdamW optimizer with a batch size of 4096 and a cosine learning rate schedule starting at $5 \times 10^{-5}$. The implementation of the pretraining is based on RedCoast (Tan et al., 2024) using bf16 mixed precision and the pretraining takes approximately 24 hours on 256 TPU-v4 cores (Jouppi et al., 2023).

### 3.2. CTCL-Topic

Another key component of CTCL is a high-quality and diverse clustering schema: a universal topic model based on document embeddings. This model is designed to identify a topic index for a given document, along with 10 representative keywords associated with the identified topic.

The topic model is used to capture high-level distributional information from the private training data. To ensure universality, the model is designed to generalize well across a wide range of downstream documents, always identifying a relevant topic. To achieve this, we constructed the topic model using Wikipedia (Foundation, 2023), a large-scale, diverse, and commonly recognized high-quality corpus.

**Topic Model Setup**   Specifically, we utilized the November 2023 version of Wikipedia, which contains over 6 million pages. A publicly available 20M-parameter document embedding model[†] was applied to the entire Wikipedia corpus, followed by HDBSCAN clustering (McInnes et al., 2017), resulting in the identification of 1,300 clusters, each treated as a distinct topic. To represent each topic, we employed the KeyBERT (Sharma & Li, 2019) to annotate 10 keywords for each cluster. The implementation of the pipeline above is based on BERTopic[‡] (Grootendorst, 2022).

### 3.3. Learning the Private Domain

When applying CTCL to the downstream private domains (shown by Part B in Figure 1), we use CTCL-Topic to

---

[†] https://huggingface.co/
sentence-transformers/all-MiniLM-L6-v2
[‡] https://maartengr.github.io/BERTopic/

| | PubMed (Medical Paper Abstract) | | | | | | | |
|---|---|---|---|---|---|---|---|---|
| | $\epsilon = \infty$ | | $\epsilon = 4$ | | $\epsilon = 2$ | | $\epsilon = 1$ | |
| Setting | BERT$_{Mini}$ | BERT$_{Small}$ | BERT$_{Mini}$ | BERT$_{Small}$ | BERT$_{Mini}$ | BERT$_{Small}$ | BERT$_{Mini}$ | BERT$_{Small}$ |
| GPT2$_{XL}$-1.5B (Upper Bound) | 39.6 | 42.9 | 37.7 | 40.5 | 37.3 | 40.2 | 36.8 | 39.7 |
| GPT2$_{XL}$-1.5B-LoRA (Upper Bound) | 39.4 | 42.5 | 34.7 | 37.7 | 34.9 | 37.9 | 34.9 | 37.9 |
| Downstream DPFT (No Synthetic Data) | **44.3** | **46.0** | 30.7 | 34.1 | 28.9 | 32.5 | 26.7 | 30.4 |
| Private Evolution (PE) (Lin et al., 2024) | 29.7 | 31.8 | 29.6 | 31.8 | 29.7 | 31.9 | 29.8 | 31.9 |
| AUG-PE + Mixtral-8x7B (Xie et al., 2024) | 24.9 | 27.6 | - | - | - | - | 24.5 | 27.1 |
| AUG-PE + GPT-3.5 (Xie et al., 2024) | 30.4 | 32.7 | 30.3 | 32.5 | 30.2 | 32.5 | 30.1 | 32.4 |
| GPT2$_{Small}$ (Yue et al., 2023) | 38.1 | 41.6 | 35.0 | 37.4 | 32.0 | 34.4 | 26.8 | 29.3 |
| GPT2$_{Small}$ + Resample (Yu et al., 2024) | 39.0 | 42.4 | 35.3 | 37.5 | 33.0 | 35.1 | 27.6 | 29.1 |
| BART$_{Base}$ (Yue et al., 2023) | 40.9 | 43.9 | 30.5 | 32.4 | 28.9 | 30.8 | 26.7 | 28.5 |
| BART$_{Base}$ + Resample (Yu et al., 2024) | 41.3 | 44.2 | 30.7 | 32.5 | 29.0 | 30.7 | 26.5 | 28.0 |
| Ours | 41.5 | 44.6 | **35.9** | **38.1** | **35.4** | **37.6** | **34.5** | **36.7** |

*Table 3.* Performance of PubMed evaluated by next-word prediction accuracy of downstream models (BERT$_{Mini}$ and BERT$_{Small}$). A smaller privacy budget ($\epsilon$) corresponds to a stricter privacy constraint. See §4.1.2 for details of different baselines.

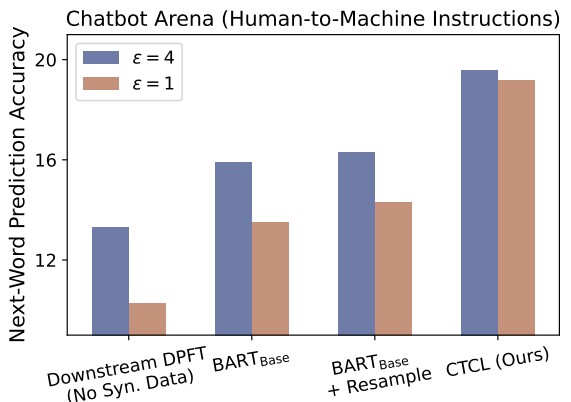
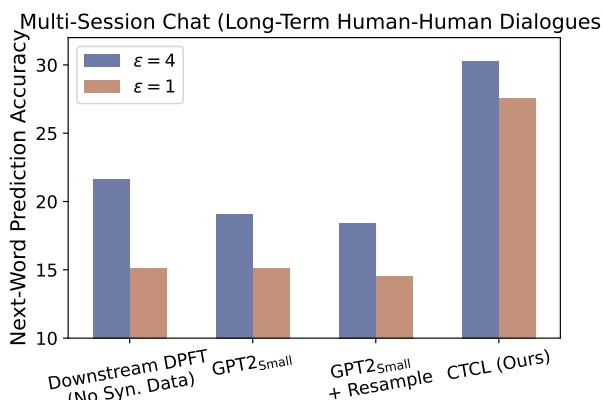

*Figure 2.* Next-word prediction accuracy of the downstream model BERT$_{Mini}$ in the *Chatbot Arena Instruction* and *Multi-Session Chat* domains. Comparing the the blue and yellow bars, our framework demonstrates greater improvements over the baselines under the stricter privacy constraint $\epsilon = 1$ compared to the setting of $\epsilon = 4$.

capture the high-level distributional information across the entire private corpus (via DP topic histogram), and adapt CTCL-Generator (via DP finetuning) to learn fine-grained text information from the private data.

**DP Topic Histogram**   Using the topic model built by CTCL-Topic, we first assign a topic to each document in the private training data, by applying the same 20M document embedding model[†] on every document and finding the closest topic embedding (among the 1,300 topic embeddings obtained from §3.2). A histogram representing the topic-wise distribution of the private corpus (i.e., the proportion of documents associated with each topic) is then constructed. Gaussian noises are added properly to every bins in the histogram, and the result is a private topic histogram.

**DP Finetuning**   After the "DP Topic Histogram" process, each document in the private dataset has been assigned to a topic. Recall that in CTCL-Topic, each topic is represented by 10 keywords. Now we transform the private dataset into the `(condition, document)` pairs, where the `condition` part consists of 10 keywords corresponding to the topic assigned to the document. This dataset is then

used to DP finetune the CTCL-Generator. Note that the `condition` part is slightly different between the constructed pretraining data in §3.1 and private finetuning data here. The pretraining data has free-form text condition (obtained from Gemma-2-2B) while the finetuning data has 10 keywords as the condition. That said, if available, additional domain-specific knowledge, such as document types, can be incorporated into the condition as well. These constructed condition-document pairs align with the pretraining condition-to-document task in §3.1. This alignment is a key to benefit from our pretraining, which enables the model to effectively learn private information while being more robust to the noise in training compared to vanilla DP finetuning (Yue et al., 2023; Kurakin et al., 2023).

### 3.4. Synthetic Data Generation

The DP finetuned CTCL-Generator is sampled to generate synthetic data based on the DP topic histogram (see Part C in Figure 1). Specifically, given the desired size of the synthetic dataset (say, $N$) and the topic proportions specified by the DP topic histogram (say, $x\%$ for Topic 1, $y\%$ for Topic 2, etc), we know the number of target samples for

each topic (i.e., $xN$ for Topic 1, $yN$ for Topic 2, etc). For each topic, we use the corresponding 10 keywords as input to the DP finetuned CTCL-Generator to generate data.

# 4. Experiments

## 4.1. Experimental Setup

### 4.1.1. DOWNSTREAM TASKS

Our experiments contain three generative tasks and two classification tasks[§]. The downstream generative tasks are evaluated by the next-word prediction accuracy, which needs the synthetic data to preserve fine-grained textual information from the private data. In contrast, the downstream classification tasks usually rely on co-occurrence patterns between labels and words in the synthetic data. Therefore, generative tasks tend to be more challenging than classification tasks.

**Generative Tasks** Three generative downstream tasks are chosen to cover a diverse set of the practical scenarios. Specifically, we include *PubMed* (Yu et al., 2023) to represent the academic medical domain, *Chatbot Arena* (Zheng et al., 2023) for human-to-machine interactions, and *Multi-Session Chat* (Xu, 2021) for human-to-human everyday dialogues. Following the evaluation setup in (Xie et al., 2024), we train 10M-level downstream causal LMs on the synthetic datasets, and use next-word prediction accuracy on the real test data as the primary quality metric.

**Classification Tasks** We conduct experiments on two classification tasks: Yelp (Yelp, Inc.) and OpenReview (Xie et al., 2024), both of which are 5-way classification, with Yelp focusing on business reviews and OpenReview on academic paper reviews. The performance is measured by the accuracy of a downstream classifier trained on the synthetic data.

To mitigate concerns regarding data contamination, we use a search engine (Liu et al., 2024) indexed on RedPajama (Computer, 2023) (a superset of our pretraining corpus) to identify potential overlaps between our downstream and pretraining data. Our analysis detects no overlap between our training data and the five downstream datasets. Additionally, for the PubMed dataset, all included samples are dated within August 2023, ensuring they were published after the release of our pretraining corpus in June 2023.

### 4.1.2. BASELINES

**Direct DP Finetuning Downstream Models** A straightforward approach to obtain a downstream model is to directly perform DP finetuning of the downstream model on the private data, without using the synthetic data. For simplicity, we refer to this baseline as "Downstream DPFT" throughout this paper.

**Vanilla DP Finetuning** We conduct standard DP finetuning (Yue et al., 2023) on BART_{Base} (Lewis, 2019) and GPT2_{Small} (Radford et al., 2019), both of which have comparable O(100M) model sizes as that of the generator in our framework. Additionally, we include DP finetuning of GPT2_{XL}-1.5B (Radford et al., 2019) as an upper bound. Given prior findings that LoRA finetuning can outperform full-model finetuning under DP constraints (Kurakin et al., 2023), we also evaluate a LoRA DP-finetuned variant of GPT2_{XL}-1.5B as another upper bound. Notably, while LoRA reduces trainable parameters, it does not significantly decrease resource demands since backpropagation is still required through the full backbone LLM.

**Post-Generation Resampling** Yu et al. (2024) proposes to refine the synthesizd dataset by a resampling technique, in order to better align with statistical properties derived from the private data.

**Private Evolution (PE)** We include results from the original PE (Lin et al., 2024) and its augmented variant, AUG-PE (Xie et al., 2024), as the examplar of LLM prompting based data synthesis approach.

### 4.1.3. HYPERPARAMETERS

**DP Finetuning and Sample Generation** For all settings involving DP finetuning, we use DP-Adam for 2000 steps with a batch size of 4096, a gradient norm clip of 1.0, and a weight decay of 0.1. The learning rate follows a linear decay schedule with 100 warmup steps, and the peak learning rate is selected from the range $[1, 4] \times [10^{-3}, 10^{-4}, 10^{-5}]$ based on validation performance. The privacy budget accounts for both DP model finetuning and the collection of DP topic histogram statistics. We apply a Gaussian noise multiplier of 10 to the DP topic histogram. The noise multipliers for DP finetuning vary across settings depending on the training data size and the presence of a topic histogram [¶]. For the sample generation process, we generate 400K synthetic examples using nucleus sampling with top-p = 0.95 and a maximum sequence length of 512 tokens. For upper-bound experiments with GPT2_{XL}-1.5B, we reduce the batch size to 256 to mitigate computational costs. The implementation of DP finetuning is based on RedCoast (Tan et al., 2024) using full fp32 precision.

**Downstream Model Training and Evaluation** We follow the evaluation of (Xie et al., 2024) for both generative and classification tasks. For generative tasks, we train the causal versions of BERT_{Mini} and BERT_{Small} using a linear learning rate schedule from 0.0003 to 0, a batch size of 64, and a total of 6000 steps, with a weight decay of 0.01. For classification tasks, we finetune a RoBERTa-base model under the same hyperparameter settings as in generative tasks above, except for a learning rate of $3 \times 10^{-5}$.

---

§Training data sizes can be found in Appendix C.

¶More details of noise multipliers and privacy budget allocation can be found in Appendix E and D.

| Setting | Yelp | | | | OpenReview | | | |
|---|---|---|---|---|---|---|---|---|
| | $\epsilon = \infty$ | $\epsilon = 4$ | $\epsilon = 2$ | $\epsilon = 1$ | $\epsilon = \infty$ | $\epsilon = 4$ | $\epsilon = 2$ | $\epsilon = 1$ |
| GPT2$_{XL}$-1.5B (Upper Bound) | 71.1 | 69.4 | 68.2 | 68.2 | 49.1 | 46.6 | 46.3 | 45.4 |
| GPT2$_{XL}$-1.5B-LoRA (Upper Bound) | 70.3 | 67.7 | 67.6 | 67.7 | 51.0 | 46.2 | 45.3 | 46.0 |
| Downstream DPFT (No Synthetic Data) | **76.0** | 67.5 | 67.2 | 66.8 | 50.8 | 32.0 | 32.0 | 32.0 |
| Private Evolution (PE) (Lin et al., 2024) | 67.9 | 67.1 | 67.2 | 67.6 | 42.4 | 43.5 | 43.7 | 42.9 |
| AUG-PE (Xie et al., 2024) | 68.4 | 68.1 | 67.8 | **67.9** | 43.5 | 44.6 | 44.5 | 43.1 |
| GPT2$_{Small}$ (Yue et al., 2023) | 71.0 | **68.2** | 67.9 | **67.9** | 52.1 | 41.1 | 38.5 | 35.1 |
| BART$_{Base}$ (Yue et al., 2023) | 70.7 | 66.3 | 66.9 | 66.9 | 52.6 | 44.7 | 42.2 | 25.7 |
| Ours | 70.5 | 68.1 | **68.0** | 67.7 | **53.9** | **46.5** | **47.1** | **46.2** |

*Table 4.* Accuracy of downstream models in the classification tasks. A smaller privacy budget ($\epsilon$) corresponds to a stricter privacy constraint. See §4.1.2 for details of different baselines.

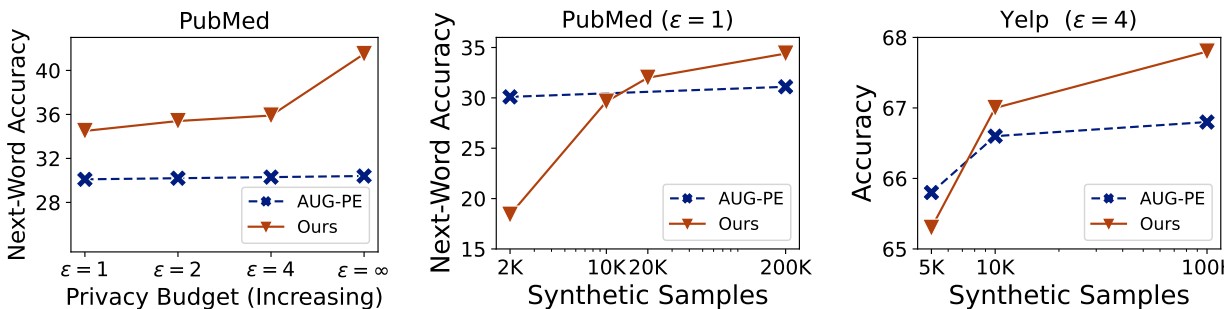

*Figure 3.* Scalability investigation results. The x-axes represent the increasing privacy budget or the number of synthetic examples, while the y-axes indicate the performance of downstream models trained on synthetic data.

## 4.2. Results

### 4.2.1. GENERATIVE TASKS

Table 3 and Figure 2 present the results of three generative tasks ‖. Our framework consistently outperforms baselines under different DP constraints and achieves performance close to the upper bounds. Moreover, as shown in Figure 2, the performance gap between our framework and the baselines widens under tighter privacy constraints (i.e., comparing the patterns of blue and yellow bars), highlighting its robustness. This can be attributed to our framework's ability to simultaneously learn both high-level and fine-grained information from private data.

Our results also reflect the limitations of the baselines. Specifically, when there is no DP constraint, direct downstream finetuning on the real data (without synthetic data) achieves the best performance across all three tasks. However, adding DP training noise leads substantial performance drop, indicating the vulnerability towards DP noise of small downstream models. Additionally, the performance of PE methods (Xie et al., 2024; Lin et al., 2024) remains almost unchanged across different privacy constraints, which also indicates that these methods do not fully exploit the increased privacy budget. This limitation may stem from their constrained capacity (i.e., only via the nearest neighbors)

to effectively capture information in the private data. Moreover, a comparison of different LMs within the AUG-PE framework reveals a significant performance gap between GPT-3.5 API and the open-source Mistral-8x7B, despite the latter also being considered a strong model. This suggests that the effectiveness of PE methods heavily relies on the exceptional capacity and creativity of the backbone LLM.

### 4.2.2. CLASSIFICATION TASKS

As shown in Table 4, our model still achieves performance that is either superior to or on par with the best-performing methods. PE-based approaches demonstrate stronger results in classification tasks compared to their performance on generative tasks. This may be because that classification primarily relies on synthetic data to capture associations between labels and specific words or phrases, which is an objective that PE methods can effectively achieve by prompting LLMs properly. In contrast, generative tasks require a deeper resemble of finer-grained textual information from private data, which poses greater challenges for PE methods.

## 4.3. Analysis and Ablation Study

### 4.3.1. SCALABILITY

The privacy budget and the size of the generated synthetic data are two key factors influencing the performance of data synthesis. In this study, we examine the effect of these

---
‖A complete result table is available in Appendix K.

| Setting | $\epsilon = \infty$ | $\epsilon = 1$ |
|---|---|---|
| BART$_{Base}$ | 63.1 | 572.9 |
| BART$_{Base}$ + Keywords | 49.3 | 291.6 |
| BART$_{Base}$ + Keywords + Pretraining (Ours) | 48.4 | 125.6 |

*Table 5.* Ablation study results evaluated by the downstream language model's perplexity (lower values indicate better performance). A privacy budget of $\epsilon = \infty$ means no DP training noise during the finetuning of the data generator.

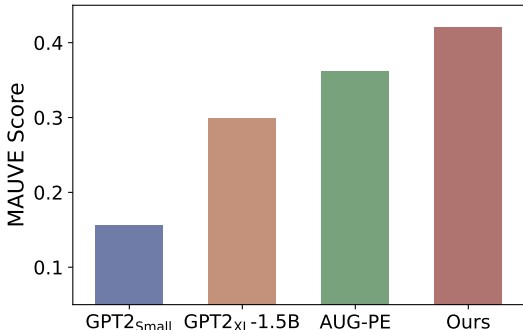

*Figure 4.* MAUVE scores on PubMed dataset under $\epsilon = 4$. Note that only the relative rankings instead of the absolute scores matters here, because the score scale can change a lot with slightly different evaluation configurations (e.g., use a different embedding model).

factors, focusing on a comparative analysis between our framework and AUG-PE, an exemplar prompting-based approach. To investigate the impact of synthetic data size, we follow the experimental setup of (Xie et al., 2024) and extend AUG-PE's sample sizes to 200K for the PubMed dataset and 100K for the Yelp dataset. The PubMed expansion is achieved by combining two runs of data synthesis using GPT-3.5 and Llama-3-8b-Instruct, while the Yelp expansion uses GPT2-Large as reported in (Xie et al., 2024).

Regarding privacy budget scalability, as illustrated in the leftmost plot of Figure 3 and briefly discussed in §4.2.1, AUG-PE does not benefit from an increased privacy budget, whereas our framework continues to improve under the same conditions. For synthetic data size, the second and third plots in Figure 3 show that when the number of synthetic examples remains in the thousand-level range, AUG-PE produces higher-quality datasets. However, its performance plateaus beyond 10K examples. In contrast, our framework exhibits continuous improvement as the dataset size increases. These findings align with our discussion in §2 and Appendix B on the size limitations of PE method.

Overall, our approach demonstrates superior scalability in terms of both privacy budget and synthetic data size.

### 4.3.2. ABLATION STUDY

In this study, we validate the importance of two key components in our framework: 1) pretraining the generator and 2) incorporating keyword-based conditions during DP fine-

**BART-Base (Downstream Model Performance: 30.5%):** We explored the relationship between molecular interaction, NCT-2 and NCT-3 (NCT-1), NCT-4, NCT and NCT–1.5 (NCT), NCT-, NCT-6, and NCT[4]. In a recent clinical trial, we described an enzyme in NCT-10 that enabled novel processes to explore novel approaches for NCT- 2.3 to NCT-III. [...]

**GPT2$_{XL}$-1.5B (Upper Bound, Downstream Model Performance: 37.7%)** The ability of leptin to induce weight loss, to stimulate ectothermic thermogenesis, and to augment activity of the AMPK system and the AMPK-dependent lipoprotein lipase activity, was examined. Circulating concentrations of leptin were assessed in the femoral adipose fat pad of the lean and obese [...]

**AUG-PE + GPT-3.5 (Downstream Model Performance: 30.3%)** An increasing incidence of aneurysmal subarachnoid hemorrhage (SAH) remains high, necessitating prompt intervention. The recognition and treatment options, including both surgical and endovascular approaches, have emerged as key components of tertiary management. [...]

**Ours (Downstream Model Performance: 35.9%):** To develop a therapeutic formula to reduce rates of morbidity that occur in people with a combination of cardiovascular problems. We used a multi-state, multidisciplinary approach to the research of the clinical manifestations of cardiovascular problems with the introduction of a biocontrol. [...]

*Table 6.* Synthetic data samples on PubMed under $\epsilon = 4$. Randomly Sampled. Obvious disfluent cuts are highlighted in red.

tuning. Specifically, starting from standard DP finetuning, we sequentially introduce these components and measure the downstream model's perplexity. The results, presented in Table 5, demonstrate the following: first, a comparison between "BART$_{Base}$" and "BART$_{Base}$ + Keywords" reveals that incorporating keywords during finetuning significantly improves performance, regardless of the presence of DP training noise. Second, a comparison between "BART$_{Base}$ + Keywords" and "BART$_{Base}$ + Keywords + Pretraining" indicates that pretraining offers limited benefits in noise-free settings but provides a clear advantage when the DP training noise is added.

We also compute the MAUVE score (Pillutla et al., 2021) to measure the distribution similarity between the generated synthetic data and the PubMed test set. As shown in Figure 4, our method achieves the highest MAUVE score among the compared methods, showing the effectiveness of our topic-wise distribution alignment during the generation process (§3.1). Moreover, a comparison between GPT2$_{XL}$-1.5B and GPT2$_{Small}$ reveals that DP finetuning on a larger model better captures high-level distributional patterns. Furthermore, we find that the high-level distribution similarity measured by MAUVE is not the sole determinant of synthetic data quality. For instance, while the synthetic data from GPT2$_{XL}$-1.5B has a lower MAUVE score than that of our approach, the model trained on it achieves a higher downstream performance (37.7% vs. 35.9%) in Table 3.

### 4.3.3. Synthetic Samples

Table 6 presents synthetic samples generated by our framework and several baselines. Under the DP training noise, the BART$_{Base}$ model tends to produce repetitive content. In contrast, our framework, built on the same lightweight model architecture, maintains the sentence fluency well. Interestingly, while the AUG-PE method generates fluent sentences using the powerful GPT-3.5, its downstream performance is only comparable to that of the DP-finetuned BART$_{Base}$. This suggests that in the context of data synthesis, the quality of the surface form (e.g., fluency and coherence) may not be the most critical factor. Generating synthetic data that is useful for the downstream model development is more important than generating fluent data.

## 5. Conclusion

In this work, we propose a novel framework CTCL for synthesizing private domain data, which integrates a universal topic model with a lightweight 140M conditional language model. This framework captures both high-level, topic-specific information and fine-grained, context-sensitive details of the private domain in a modular and efficient manner. Through evaluations across five diverse downstream domains, we demonstrate that the synthesized data generated by CTCL outperforms baseline methods, including vanilla differential privacy finetuning and prompting-based approaches such as private evolution.

While the proposed CTCL framework outperforms existing privacy-preserving methods in the downstream tasks, one limitation is that the synthetic data generated by our 140M-parameter still lack the fluency and coherence compared to those generated by prompting the latest billion-size LLMs. How to leverage LLMs for data synthesis in the resource-constrained setting (the setting which this paper focuses on), would be one important area for future exploration.

Another interesting direction for future work is to generalize our framework to the multi-modal settings, e.g., data synthesis for the vision-language tasks. This requires appropriate architectural adaptations, e.g., by replacing language models with diffusion-based models for image generation.

## Impact Statement

This paper presents work whose goal is to advance the field of Machine Learning. There are many potential societal consequences of our work, none which we feel must be specifically highlighted here.

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

# A. Background on Differential Privacy

We use the standard $(\epsilon, \delta)$-differential privacy (DP) guarantee (Dwork et al., 2006) to measure the privacy risk of an ML algorithm memorizing individual records in the sensitive training data. For simplicity, we present a brief description below, and defer to (Dwork et al., 2014; Ponomareva et al., 2023) for more details.

**Definition A.1** $((\epsilon, \delta)$-DP (Dwork et al., 2006)). A randomized algorithm $\mathcal{M}$ satisfies $(\epsilon, \delta)$-DP if for any two neighboring datasets $\mathbb{D}$, $\mathbb{D}'$ (defined by adding or removing one record from the dataset), and for any $\mathcal{S} \subseteq \mathrm{Range}(\mathcal{M})$, where $\mathrm{Range}(\mathcal{M})$ is the set of all outcomes of $\mathcal{M}$:

$$\Pr[\mathcal{M}(\mathbb{D}) \in \mathcal{S}] \leq e^{\epsilon} \Pr[\mathcal{M}(\mathbb{D}') \in \mathcal{S}] + \delta.$$

At the high level, $(\epsilon, \delta)$-DP provides a formal definition that adding or removing a single record from the dataset should not have a large influence on the algorithm output. This indicates that the algorithm only learns the common knowledge from the entire dataset.

To help readers better understand our paper, we now briefly describe two important facts about $(\epsilon, \delta)$-DP. First, one popular approach to DP training is DP-SGD (Song et al., 2013; Bassily et al., 2014; Abadi et al., 2016) or variants such as DP-Adam (Li et al., 2022; Yu et al., 2022), which modifies the standard SGD algorithm by clipping per-sample gradients and adding noise to each gradient updates during training. We use DP-Adam to train LMs throughout this paper. Second, any post-processing of a private algorithm's output cannot make it less private (Dwork et al., 2014). In our case, this property means that the synthetic dataset generated by a DP finetuned LM has the same $(\epsilon, \delta)$-DP guarantee as that of the DP finetuned LM.

# B. Supplementary Discussion on Related Work

This paper focuses on generating privacy-preserving text data that resemble a private data source. We discuss more about prior work here in addition to §2.

**Synthetic text data generated by DP-finetuned LMs.** This is a popular approach: an LM is first finetuned on the private data with DP, and then sampled to generate synthetic data (Bommasani et al., 2019; Putta et al., 2022; Mattern et al., 2022a; Yue et al., 2023; Kurakin et al., 2023; Yu et al., 2024; Wang et al., 2024; Ochs & Habernal, 2024; Carranza et al., 2024). While this paper follows a similar approach, we primarily focus on improving the data generation quality from a small O(100M)-scale LM. By carefully finetuning on the public and private data, the synthetic data generated by our method have significantly better quality than that obtained by the previous DP finetuning approaches for the O(100M)-scale LMs.

Existing DP finetuning-based approaches for synthetic data generation often rely on the strong capability and generalability of billion-scale LLMs, especially when the downstream tasks are the challenging generative tasks as opposed to the simpler classification tasks. For instance, (Yu et al., 2024) DP finetune LLaMA-7B to synthesize Chatbot Arena-style short (often one-sentence) human-to-machine instructions. While CTCL also incorporates a DP finetuning step on private data, it significantly reduces the computational cost by using a backbone LM with only 140M parameters, making it much more acceptable for real-world resource-constrained applications.

**Synthetic text data that only require LLM API access.** Because DP finetuning can be expensive and sometimes impossible (e.g., for non-public models), this line of work explores data synthesis assuming only access to the LLM inference APIs. Simply relying on the high-level knowledge about a private domain to design proper LLM prompts is not enough to generate synthetic data that well represent the actual private domain (Wu et al., 2024a). Using a small DP language model to filter the initial synthetic data makes the data more similar to private data, but does not close the distribution gap for training a model for the downstream task in the private domain (Wu et al., 2024a; Zhang et al., 2025).

The Private Evolution (PE) framework, initially developed by Lin et al. (2024) for the image domain, and later extended to the text domain in (Xie et al., 2024; Hou et al., 2024), proposes to "refine" (i.e., select good examples from) the current synthetic dataset according to the closeness of each example with respect to the private dataset. A similar idea is also explored by Zhao et al. (2024). The key idea behind PE is to measure closeness using DP nearest neighbor histogram: if an example is close to the private distribution, then it would receive a lot nearest neighbor votes from the private examples. PE starts with an initial dataset generated by the state-of-the-art LLMs via API access (using domain knowledge to design LLM prompts). PE then works by iteratively selecting good examples (measured by the DP nearest neighbor histogram) and

using LLMs to generate more similar examples. In our experiments, we show that PE performs worse than DP finetuning especially for generative tasks, even when we are only allowed to finetune a small (million parameters) LMs.

Unlike prompting-based approaches such as PE, CTCL does not depend on prompting advanced LLMs or external APIs when applied on downstream data. Although a prompting step is involved during CTCL's pretraining phase in §3.1 (note that this step only needs to be done once), our prompt is only designed for summarizing a public document, without requiring the strong creativity capabilities from advanced LLMs. In practice, for this step, we only need a lightweight 2B-parameter LLM, ensuring that the constructed data can be scaled up to pretraining level. Table 1 presents the examples of prompts used in existing prompting-based data synthesis approaches as well as the one used in our pretraining data construction.

The size of the synthetic dataset in PE-based approaches is often constrained due to the sample-wise noise introduced by the differential-private nearest neighbor (DP-NN) process. In this setup, DP-NN serves as the only mechanism for incorporating the information from the private data into the synthetic dataset. Specifically, DP-NN identifies the nearest neighbors of each private data sample within the synthetic dataset. To preserve privacy, DP noise is added to each synthetic sample. In this process, each synthetic sample acts as a bin, with its count indicating how many private data samples identify it as their nearest neighbor. Noise is then applied to these counts to ensure privacy. However, this process requires each bin to contain a sufficiently large number of counts; otherwise, the noise overwhelms the signal, making it difficult to distinguish between zero and small counts. Consequently, the synthetic dataset size must be limited, as an excessive number of bins would make DP-NN ineffective. To overcome this limitation, a variation process is often employed, where LLM APIs are prompted to generate additional samples based on the initially selected subset. However, since the private data does not directly influence this generation process, the final synthetic dataset theoretically contains no more information from the private data than the small subset originally selected. A few concurrent work (Hou et al., 2025; Zou et al., 2025) further improved the PE algorithm; while we did not directly compare to their experimental results, a lot of our discussion can still apply to these improved PE methods.

Nagesh et al. (2024) propose an approach that privately learns the probability distribution of keyphrases via kernel density estimation, followed by sampling sequences of keyphrases to seed LLM prompts. To better capture the correlations between the sampled keyphrases, this method requires estimating the distribution of keyphrases at varying lengths. As a result, it is hard to scale this method to generating very long documents.

Another line of work explores generating private-preserving few-shot examples for in-context learning (ICL), e.g., (Tang et al., 2024; Wu et al., 2024b; Duan et al., 2023). Given a reasonable DP guarantee, these methods can only generate a limited amount of synthetic data, e.g., a few for ICL, or a few thousands (Amin et al., 2024). By contrast, our method (based on DP finetuning) can generate a much larger dataset of synthetic examples.

**Text privatization based on word or sentence perturbations.** These approaches, e.g., (Feyisetan et al., 2020; Mattern et al., 2022b; Carvalho et al., 2023; Utpala et al., 2023), usually use a different DP notion and perform worse than the approaches discussed above. We defer interested readers to Appendix C.11 in (Xie et al., 2024) for more details.

**Conditional generation.** In addition to privacy-preserving methods for synthesizing data for a specific domain, conditional generation is also explored to improve the diversity and generative quality of models in less privacy sensitive applications. We now discuss a few recent work in large language models. In concurrent work, Gao et al. (2025) discussed metadata conditioning for pre-trainig large language models; DeSalvo et al. (2025) developed soft prompt as condition to improve synthetic data generation.

## C. Dataset Sizes

| Dataset | Train | Valid | Test |
|---|---|---|---|
| PubMed | 75,316 | 14,423 | 4,453 |
| Chatbot Arena | 180,000 | 5,000 | 3,819 |
| Multi-Session Chat | 17,940 | 3,000 | 2,505 |
| Yelp | 1,939,290 | 5,000 | 5,000 |
| OpenReview | 8,396 | 2,798 | 2,798 |

*Table 7.* Sizes of datasets in our experiments.

## D. Privacy Budget Allocation

Our privacy budget allocation follows the "Post-Generation Resample" algorithm proposed by Yu et al. (2024), which also has the DP-histogram and DP-finetune components. For the DP-histogram step, we adopt the same configuration—adding a Gaussian noise on every histogram bin with a standard deviation of 10. The overall privacy budget ($\epsilon$) is computed as the composition of both steps using the standard DP accounting library**.

To illustrate, we provide the individual $\epsilon$ values in the table below for the DP-histogram and DP-finetuning steps on the PubMed dataset (75K training samples), under composed privacy budgets of $\epsilon = 4$, 2, and 1, and $\delta =$. As shown below, the DP-histogram step consistently accounts for a small portion of the overall budget—for instance, $\epsilon = 0.39$ for DP-histogram versus $\epsilon = 3.96$ for DP-finetuning when the composed $\epsilon$ is 4. This trend aligns with the observations in (Yu et al., 2024) and highlights that our CTCL-Topic design achieves strong performance while consuming only a small portion of the privacy budget.

| $\epsilon$ (Composed) | $\epsilon$ (DP-Histogram) | $\epsilon$ (DP-Finetune) |
|---|---|---|
| 4 | 0.39 | 3.96 |
| 2 | 0.39 | 1.94 |
| 1 | 0.39 | 0.9 |

*Table 8.* The allocation of provacy budget between DP-Histogram and DP-Finetune.

Additionally, we find that changing the privacy allocation to the DP-histogram step only has very small impact to the DP-finetuning step. For example, in the $\epsilon = 4$ setting on PubMed, increasing the Gaussian noise in the histogram step from 10 to 20 alters the DP-finetuning $\epsilon$ marginally—from 3.96 to 3.99. This corresponds to a small change in the noise multiplier for DP-finetuning (from 3.03 to 3.02).

---

**https://github.com/google/differential-privacy/tree/main/python/dp_accounting/dp_accounting/pld

# E. Noise Multipliers

The table below gives the noise multipliers used when DP finetuning LMs in our experiments. Following (Yue et al., 2023), we set $\delta = \frac{1}{N \log N}$, where $N$ is the size of private training set. Given a desired $(\epsilon, \delta)$-DP guarantee, the noise multipliers are computed using the standard *dp_accounting* package (DP Team, 2022). As pointed out in Appendix A, we use DP-Adam for DP finetuning and follow the standard Gaussian mechanism to obtain $(\epsilon, \delta)$-DP guarantee. Compared to the "Vanilla DP Finetune" approach, the noise multiplier used by our method is slightly larger, because we need to allocate a small portion of the privacy budget to the DP topic histogram (see Appendix D). Besides, GPT2$_{XL}$-1.5B has much smaller noise multipliers because we reduce the training batch size from 4096 to 256 to save computational resources. For other non-DP training hyperparameters, see §4.1.3.

| | $\epsilon = \infty$ | $\epsilon = 4$ | $\epsilon = 2$ | $\epsilon = 1$ |
|---|---|---|---|---|
| **PubMed** | | | | |
| Vanilla DP Finetune (BART$_{Base}$ and GPT2$_{Small}$) | 0 | 3.01 | 5.49 | 10.3 |
| GPT2$_{XL}$-1.5B (reduced batch size) | 0 | 0.63 | 0.77 | 0.97 |
| Ours | 0 | 3.03 | 5.63 | 11.33 |
| **Chatbot Arena** | | | | |
| Vanilla DP Finetune (BART$_{Base}$ and GPT2$_{Small}$) | 0 | 1.47 | 2.5 | 4.58 |
| GPT2$_{XL}$-1.5B (reduced batch size) | 0 | 0.56 | 0.67 | 0.78 |
| Ours | 0 | 1.48 | 2.57 | 5.08 |
| **Multi-Session Chat** | | | | |
| Vanilla DP Finetune (BART$_{Base}$ and GPT2$_{Small}$) | 0 | 11.38 | 21.01 | 39.41 |
| GPT2$_{XL}$-1.5B (reduced batch size) | 0 | 1.00 | 1.52 | 2.61 |
| Ours | 0 | 11.45 | 21.47 | 42.70 |
| **Yelp** | | | | |
| Vanilla DP Finetune (BART$_{Base}$ and GPT2$_{Small}$) | 0 | 0.63 | 0.77 | 0.91 |
| GPT2$_{XL}$-1.5B (reduced batch size) | 0 | 0.51 | 0.6 | 0.69 |
| Ours | 0 | 0.63 | 0.77 | 0.94 |
| **OpenReview** | | | | |
| Vanilla DP Finetune (BART$_{Base}$ and GPT2$_{Small}$) | 0 | 23.3 | 42.87 | 80.05 |
| GPT2$_{XL}$-1.5B (reduced batch size) | 0 | 1.64 | 2.81 | 5.11 |
| Ours | 0 | 23.44 | 43.72 | 86.05 |

*Table 9.* The noise multipliers used when DP finetuning LMs in our experiments.

## F. Topic Model's Coverage

We choose Wikipedia as the basis for our topic model because it is large-scale, high-quality, and semantically diverse. Most real-world documents with meaningful content would fall within Wikipedia's domain. In practice, it is rare to encounter texts that are entirely outside of its scope. To illustrate the coverage of our topic model, the table below presents topics sampled from five diverse datasets used in our experiments. These examples demonstrate that the model captures a broad range of universal topics:

| Dataset | Sample topic 1 | Sample Topic 2 |
|---|---|---|
| PubMed | microscopy/electron/... | rna/mir/gene/... |
| Chatbot Arena | gameplay/rpg/wii/... | eruptions/volcano/... |
| Multi-Session Chat | comedian/presenter/... | oscar/nominations/... |
| Yelp | restaurant/diner/... | theater/cinemas/... |
| OpenReview | grammar/syntax/... | computational/turing/... |

*Table 10.* Example topics in the datasets of our experiments.

The "out-of-domain" text is also considered and processed in our experiments. Specifically, in our topic model, there is an "unclassified" bin for documents that are not sufficiently close to any learned topic. The table below reports the number of unclassified samples in each dataset, along with example such texts:

| Dataset | Training size | Unclassified samples | Unclassified sample |
|---|---|---|---|
| PubMed | 75,316 | 0 | |
| Chatbot Arena | 180,000 | 86 | hello. i have no idea who am i, maybe you can help? |
| Multi-Session Chat | 17,940 | 0 | |
| Yelp | 1,939,290 | 4 | I am unable to provide check number. |
| OpenReview | 8,396 | 0 | |

*Table 11.* Analysis of unclassified samples in every datasets.

As shown in Table 11, the number of unclassified samples is neglectable across all datasets. Furthermore, those that do fall into the unclassified bin typically contain little to no substantive content, confirming the broad coverage of our topic model.

## G. Computational Cost and Tradeoffs

Our pretraining pipeline involves three main components:

- Data Curation: Running inference using a 2B LLM on a 430M pretraining documents.
- Generator Pretraining: Training a 140M conditional LM on 430M (condition, document) pairs.
- Topic Modeling: Generating embeddings for 6M Wikipedia documents with a 20M document embedding model, followed by a HDBSCAN clustering.

Among these, data curation for pretraining dominates the computational cost due to the scale of LLM inference. This remains lighter than or comparable to the cost of the pretraining of a 2B LLM. Note that after we release our pretrained CTCL-Generator and CTCL-Topic, other uses can skip the pretraining step (i.e., do not need to pay the pretraining costs), and directly finetune the model on their private data.

For the synthetic data generation stage, our approach samples from a small 140M model after finetuning on the private data. By contrast, existing PE-based methods often involve significant API costs that are proportionally to the sample size. For instance, in the experiments of AUG-PE (Xie et al., 2024), they synthesize 2,000 samples using $T = 10$ evolution iterations, each with $L = 4$ variations, totaling $80,000$ $(2,000 \times 10 \times 4)$ ChatGPT requests.

# H. Pseudo-code

We provide pseudo-code of our framework corresponding to Section 3 and Figure 1.

---

**Algorithm 1** A. Pretraining with Public Corpora

---

**CTCL-Topic Development**
  1: Cluster Wikipedia document embeddings via HDBSCAN.
  2: Assign top-10 keywords to each cluster as topics.

**CTCL-Generator Pretraining**
  1: Extract aspects (e.g., keywords, document type) from public corpora such as Slimpajama.
  2: Train a 140M-parameter conditional LM on (aspect, document) pairs.

---

---

**Algorithm 2** B. Learning the Private Domain

---

**Private Topic Histogram Construction**
  1: Use CTCL-Topic to assign topics to each private document.
  2: Construct a DP histogram to represent topic-level distribution of the private domain.

**DP Finetuning CTCL-Generator**
  1: Form (keyword, document) training pairs using CTCL-Topic.
  2: Finetune the CTCL-Generator using these pairs with differential privacy.

---

---

**Algorithm 3** C. Data Synthesis

---

  1: For each topic, determine the number samples based on DP topic histogram.
  2: Use the finetuned CTCL-Generator to generate documents conditioned on topic keywords.

---

# I. Discrepancy in Constructed Conditions between Pretraining and Finetuning

In our pretraining stage (§3.1), the condition is extracted for each document from Gemma 2B, while in the learning of private data, the condition is from the topic model. The distinction between pretraining and finetuning conditions is intentional and serves two main purposes: enhancing model generalizability and adhering to privacy constraints. During the pretraining stage, using the 2B LLM to extract aspects is for increasing the diversity and flexibility of the aspects the model can handle, enabling stronger controllability at inference time. Otherwise, "keywords" becomes the only aspect in the pretraining.

On the other hand, not also using Gemma 2B in the finetuning time is for privacy consideration. Specifically, if Gemma 2B were applied directly to private data, the extracted conditions would be in free-text format. This raises our concerns about sending the private information to the synthetic data generation process. In contrast, our framework ensures differential privacy by using a noised, topic-wise histogram instead. This approach preserves key high-level topic information of the private domain while satisfying privacy requirements.

To demonstrate the value of the flexible conditioning during pretraining, consider a scenario where one has a bit prior knowledge about the downstream domain (e.g., that the documents are dialogues). Our design enables such weak supervision to be incorporated, providing a better initialization for subsequent DP-finetuning. Below are examples of model generations before any finetuning, controlled only by document type, showcasing the better initialization:

---

**Condition 1**

*Document Type: News*
Keywords: Surfing, World Championship, Young Athletes

**Generation**

The 15-year-old, who is a member of the World Surfing Association (WSA) and the World Surf League (WSL), won the World Championships for the first time in his career. [...]

**Condition 2**

*Document Type: Chat*
Keywords: Surfing, World Championship, Young Athletes

**Generation**

How do you feel about the current state of surfing in the U.S. right now?
There are a lot of great young athletes out there right now, and it's going to be a really exciting year for them. [...]

**Condition 3**

*Document Type: Blog Post*
Keywords: Surfing, World Championship, Young Athletes

**Generation**

I've been fortunate enough to have the opportunity to meet some of the most talented young athletes in the world, and I'm [...]

---

*Table 12.* The flexible design of CTCL-Generator's pretraining also intends to encourage broader and more creative uses.

## J. Qualitative Analysis

We provide synthetic data examples in addition to Table 6.

---

**BART-Base (Downstream Model Performance: 30.5%)**:
**Sample 1:** We explored the relationship between molecular interaction, NCT-2 and NCT-3 (NCT-1), NCT-4, NCT and NCT–1.5 (NCT), NCT-, NCT-6, and NCT[4]. In a recent clinical trial, we described an enzyme in NCT-10 that enabled novel processes to explore novel approaches for NCT- 2.3 to NCT-III. NCT was applied to NCT and the NCT-8 was used to explore the implications of NCT on NCT in NCT. NCT-11 was used to describe NCT-7 to NCT preparation for NCT preparation. NCT and PCT were compared with NCT-16 with NCT and [...]
**Sample 2:** The authors of this paper present a unique method for understanding the mechanism of the effects of the mechanism. It aims to investigate whether the mechanism used by the methods involved in the study was used to determine the mechanism of treatment, including the mechanism of its regulation. The analysis was conducted on an open-source basis and did not rely on the traditional methods of the primary investigation, such as the original design for the method used, which looked at the mechanisms used to [...]

---

**GPT2$_{XL}$-1.5B (Upper Bound, Downstream Model Performance: 37.7%)**
**Sample 1:** The ability of leptin to induce weight loss, to stimulate ectothermic thermogenesis, and to augment activity of the AMPK system and the AMPK-dependent lipoprotein lipase activity, was examined. Circulating concentrations of leptin were assessed in the femoral adipose fat pad of the lean and obese rats. Serum leptin levels and AMPK mRNA expression were assessed in sera obtained from lean and obese female rats fed diets enriched with fat or carbohydrate. The leptin binding to the leptin receptor [...]
**Sample 2:** Nuclear stress hormone (NTH) levels increased in patients with type 2 diabetes mellitus (T2D) following therapy with metformin [13] or insulin (e.g., lispro-Lipid, insulin-C3G or insulin-G3F) [14, 15]. NTH was not increased in patients enrolled to receive metformin or lispro (or in patients taking insulin-C-based drugs (i.e., lisulin and galantangulin, alemtuhusin) [16], suggesting the insulin treatment itself might cause patients to become insulin sensitive (i.g., via altered insulin action). [...]

---

**AUG-PE + GPT-3.5 (Downstream Model Performance: 30.3%)**
**Sample 1:** Deep learning, in combination with artificial intelligence, has emerged as a powerful tool in the field of medical imaging. This study explores the application of deep learning algorithms in the identification of various medical conditions. By leveraging neural networks, this approach provides unparalleled accuracy across different modalities, enhancing our understanding of pathological processes. The integration of deep learning in medical imaging has the potential to revolutionize diagnostic processes [...]
**Sample 2:** Investigating synaptic plasticity in the context of neurological networks investigation, a cohort of patients with neurodegenerative disorders was closely monitored. The efficiency of synaptic transmission was evaluated, considering the underlying mechanisms particularly focusing on the composition of scaffolding proteins. Insights into the impact of adipocyte signaling on synaptic function were uncovered. The research further characterized the dynamics of synaptic plasticity and illuminated how [...]

---

**Ours (Downstream Model Performance: 35.9%)**:
**Sample 1:** To develop a therapeutic formula to reduce rates of morbidity that occur in people with a combination of cardiovascular problems. We used a multi-state, multidisciplinary approach to the research of the clinical manifestations of cardiovascular problems with the introduction of a biocontrol. Using a multi-choice, multi-pronged approach we evaluated a combination of the three major elements of cardiovascular problems: chronic diseases; vascular disease that occurs in chronic conditions in chronic [...]
**Sample 2:** This study describes the mechanism of action of different mechanisms to be performed to evaluate the efficacy and quality of control activities at three different levels. A cross-hatch method was used to investigate the efficacy and health outcomes of different mechanisms in all three levels of intervention. The outcome was compared with the outcomes obtained by a crosshatch method. A total of 20 control methods used during this study included 7 standard I/A for four trials. All intervention levels [...]

---

*Table 13.* Additional synthetic data samples on PubMed under $\epsilon = 4$. Randomly Sampled.

# K. Full Results of Generative Tasks

| PubMed (Medical Paper Abstract) | | | | | | | | |
|---|---|---|---|---|---|---|---|---|
| Setting | $\epsilon = \infty$ | | $\epsilon = 4$ | | $\epsilon = 2$ | | $\epsilon = 1$ | |
| | BERT$_{Mini}$ | BERT$_{Small}$ | BERT$_{Mini}$ | BERT$_{Small}$ | BERT$_{Mini}$ | BERT$_{Small}$ | BERT$_{Mini}$ | BERT$_{Small}$ |
| GPT2$_{XL}$-1.5B (Upper Bound) | 39.6 | 42.9 | 37.7 | 40.5 | 37.3 | 40.2 | 36.8 | 39.7 |
| GPT2$_{XL}$-1.5B-LoRA (Upper Bound) | 39.4 | 42.5 | 34.7 | 37.7 | 34.9 | 37.9 | 34.9 | 37.9 |
| Downstream DPFT (No Syn. Data) | **44.3** | **46.0** | 30.7 | 34.1 | 28.9 | 32.5 | 26.7 | 30.4 |
| Private Evolution (PE) (Lin et al., 2024) | 29.7 | 31.8 | 29.6 | 31.8 | 29.7 | 31.9 | 29.8 | 31.9 |
| AUG-PE + Mixtral-8x7B (Xie et al., 2024) | 24.9 | 27.6 | - | - | - | - | 24.5 | 27.1 |
| AUG-PE + GPT-3.5 (Xie et al., 2024) | 30.4 | 32.7 | 30.3 | 32.5 | 30.2 | 32.5 | 30.1 | 32.4 |
| GPT2$_{Small}$ (Yue et al., 2023) | 38.1 | 41.6 | 35.0 | 37.4 | 32.0 | 34.4 | 26.8 | 29.3 |
| GPT2$_{Small}$ + Resample (Yu et al., 2024) | 39.0 | 42.4 | 35.3 | 37.5 | 33.0 | 35.1 | 27.6 | 29.1 |
| BART$_{Base}$ (Yue et al., 2023) | 40.9 | 43.9 | 30.5 | 32.4 | 28.9 | 30.8 | 26.7 | 28.5 |
| BART$_{Base}$ + Resample (Yu et al., 2024) | 41.3 | 44.2 | 30.7 | 32.5 | 29.0 | 30.7 | 26.5 | 28.0 |
| Ours | 41.5 | 44.6 | **35.9** | **38.1** | **35.4** | **37.6** | **34.5** | **36.7** |

| Chatbot Arena (Human-to-Machine Instructions) | | | | | | | | |
|---|---|---|---|---|---|---|---|---|
| Setting | $\epsilon = \infty$ | | $\epsilon = 4$ | | $\epsilon = 2$ | | $\epsilon = 1$ | |
| | BERT$_{Mini}$ | BERT$_{Small}$ | BERT$_{Mini}$ | BERT$_{Small}$ | BERT$_{Mini}$ | BERT$_{Small}$ | BERT$_{Mini}$ | BERT$_{Small}$ |
| GPT2$_{XL}$-1.5B (Upper Bound) | 26.6 | 29.4 | 19.6 | 21.9 | 19.4 | 21.8 | 19.2 | 21.6 |
| GPT2$_{XL}$-1.5B-LoRA (Upper Bound) | 28.5 | 31.1 | 22.9 | 25 | 22.8 | 24.9 | 22.8 | 25.0 |
| Downstream DPFT (No Syn. Data) | **28.9** | **31.9** | 13.3 | 12.5 | 11.9 | 10.9 | 10.3 | 9.2 |
| GPT2$_{Small}$ (Yue et al., 2023) | 26.1 | 28.8 | 18.8 | 20.7 | 17.7 | 19.5 | 16.0 | 17.6 |
| GPT2$_{Small}$ + Resample (Yu et al., 2024) | 26.8 | 29.3 | 18.7 | 20.0 | 17.6 | 18.6 | 15.9 | 17.1 |
| BART$_{Base}$ (Yue et al., 2023) | 21.8 | 24.1 | 15.9 | 16.8 | 14.9 | 16.1 | 13.5 | 14.5 |
| BART$_{Base}$ + Resample (Yu et al., 2024) | 23.4 | 25.6 | 16.3 | 17.4 | 15.3 | 16.7 | 14.3 | 15.1 |
| Ours | 22.5 | 24.9 | **19.6** | **21.5** | **19.4** | **21.2** | **19.2** | **20.7** |

| Multi-Session Chat (Long-Term Human-Human Conversations) | | | | | | | | |
|---|---|---|---|---|---|---|---|---|
| Setting | $\epsilon = \infty$ | | $\epsilon = 4$ | | $\epsilon = 2$ | | $\epsilon = 1$ | |
| | BERT$_{Mini}$ | BERT$_{Small}$ | BERT$_{Mini}$ | BERT$_{Small}$ | BERT$_{Mini}$ | BERT$_{Small}$ | BERT$_{Mini}$ | BERT$_{Small}$ |
| GPT2$_{XL}$-1.5B (Upper Bound) | 33.2 | 35.5 | 27.5 | 30.2 | 25.3 | 28.7 | 23.9 | 27.0 |
| GPT2$_{XL}$-1.5B-LoRA (Upper Bound) | 38.3 | 40.8 | 28.4 | 30.7 | 28.4 | 31.1 | 28.8 | 31.1 |
| Downstream DPFT (No Syn. Data) | **38.8** | **40.1** | 21.6 | 17.7 | 18.9 | 11.8 | 15.1 | 6.7 |
| GPT2$_{Small}$ (Yue et al., 2023) | 34.6 | 37.2 | 19.1 | 19.9 | 20.2 | 21.4 | 15.1 | 17.3 |
| GPT2$_{Small}$ + Resample (Yu et al., 2024) | 34.6 | 37.3 | 18.4 | 17.3 | 19.9 | 18.7 | 14.5 | 13.4 |
| BART$_{Base}$ (Yue et al., 2023) | 34.2 | 36.9 | 27.8 | 29.1 | 23.8 | 25.0 | 10.8 | 11.2 |
| BART$_{Base}$ + Resample (Yu et al., 2024) | 34.8 | 37.4 | 28.1 | 29.1 | 24.2 | 25.1 | 9.1 | 9.8 |
| Ours | 34.3 | 36.4 | **30.3** | **32.6** | **29.1** | **29.7** | **27.6** | **29.3** |

*Table 14.* Performance of generative tasks evaluated by next-word prediction accuracy of downstream models (BERT$_{Mini}$ and BERT$_{Small}$). A smaller privacy budget ($\epsilon$) corresponds to a stricter privacy constraint.

