# OpenReview forum: "Synthesizing Privacy-Preserving Text Data via Finetuning *without* Finetuning Billion-Scale LLMs"
_ICML.cc/2025/Conference — ICML 2025 poster_

### Official Review · Reviewer_qpMn · 2025-03-11

**Overall Recommendation:** 3

**Summary:**

This work proposes a synthetic data generation method with a differential privacy guarantee. It leverages a lightweight data generator and a topic model for topic clustering. This method tackles the limitation of the existing DP fine-tuning method, which is expensive and relies on large LLM, and the prompt-based methods, which heavily rely on manual prompts. The extensive experiments on both generative and classification tasks show the advantage of the proposed method.

**Claims And Evidence:**

Most claims in the work are properly justified. There are some minor points to be clarified.

> "Third, PE-based methods need to balance between data quality and synthetic data size (see discussions in §2), while our framework naturally allows for unlimited data samples using the DP finetuned generator, without additional privacy costs during generation."

This sentence mentions the limitation of data quality of PE-based methods, but it needs clarification on the quality guarantee of the proposed method.

**Essential References Not Discussed:**

N/A

**Experimental Designs Or Analyses:**

The experimental designs are proper and comprehensive. The analyses are well supported.

Similar to my concern in Method, could the author provide more details on the budget allocation of the proposed method in experiments? And what is the influence of the budget allocation?

**Methods And Evaluation Criteria:**

**Method**

The whole method is reasonable and indeed provides DP guarantees. I have some confusions to be clarified.

1. The privacy budget.

While the authors mention in Appendix A, I think a more detailed privacy analysis should be provided. Since CTCL involves two DP processes, the histogram mechanism and DP fine-tuning, and the two processes are sequential, it is crucial to discuss the budget allocation for the two processes. I am also curious if the allocation will influence the utility.

2. The choice of generator and topic model

From my perspective, the generator determines the quality of the synthetic data, and the accuracy of the topic model also influences the construction of the histogram. Could the authors provide some intuitions on the choice of the architecture and size of the generator? I guess a larger size of generator can improve the quality but introduce more computation costs for fine-tuning, so the balance is crucial. I am also a little worried if a 140M model can learn the 430M dataset well. Also, how the number of clusters for the topic model is determined, as it is a universal model that needs to accommodate unseen texts?


**Evaluation**

The evaluation is proper.

**Other Comments Or Suggestions:**

N/A

**Other Strengths And Weaknesses:**

The whole paper is well written and easy to follow.

**Questions For Authors:**

See above reviews.

**Relation To Broader Scientific Literature:**

The proposed method handles limitations of existing DP methods including DP fine-tuning on large models and prompt-based methods.

**Theoretical Claims:**

N/A

---

> ### Author Rebuttal · Authors · 2025-04-01
>
> We appreciate your encouraging remarks that our method is reasonable, our experimental design is both proper and comprehensive, and that our analyses are well-supported with most claims being appropriately justified.
>
> ### Privacy Budget Allocation
> Our budget allocation follows [1], our "Post-Generation Resample" baseline, which also has both DP-histogram and DP-finetune steps. For DP-histogram, we adopt the same -- adding a Gaussian noise 10 on every histogram item. The overall privacy budget (ε) is computed as a composition of both steps using the standard `dp_accouting` package.
>
> We also show individual ε values below on PubMed, under composed $ε=4,2,1$. DP-histogram only takes a small portion of the overall budget. This aligns with observations in [1] (page 5) and highlights our CTCL-topic design achieves strong performance while consuming only a small portion of the privacy budget.
>
> | ε (Composed) | ε (DP-Histogram) | ε (DP-Finetune) |
> |:-:|:-:|:-:|
> | 4 | 0.39 | 3.96 |
> | 2 | 0.39 | 1.94 |
> | 1 | 0.39 | 0.9 |
>
> We also find that the impact of changing the allocation is insignificant. For example, in $ε=4$ setting on PubMed, increasing histogram's noise from 10 to 20 only alters the DP-finetuning budget $ε$ marginally—from 3.96 to 3.99.
>
> [1] Privacy-Preserving Instructions for Aligning Large Language Models, ICML 2024
>
>
> ### Generator design choices
> We use a seq2seq generator because its encoder is well-suited for understanding the conditions. For size, the 140M model is to balance the efficiency with generation ability, making it practical for resource-constrained scenarios.
>
> We agree other model selections might perform better, but we do not extensively explore all designs because our intuitive choice already demonstrates strong downstream performanceas in our experiments. Moreover, same as all pretraining research, due to the cost and scale, it could be super challenging to exhaustively search over all possible configurations.
>
> We would also like to emphasize that the core contribution of our work is the overall CTCL framework. Each module within the framework remains flexible and open to further design improvements.
>
> **140M model can learn 430M data well?** The large dataset is collected based on the common intuition that larger pretraining datasets typically lead to stronger models. The goal of pretraining is not for the open-ended pretraining task, but for a model with strong controllability supporting downstream tasks. Moreover, large data even on comparatively small models is a common practice. For instance, LLaMA-3 with 3B or 8B parameters are trained on 15T tokens, where the data samples also significantly exceeds the number of model parameters.
>
> ### Topic model design choices
> We choose Wikipedia for our topic model because it is large-scale, high-quality, and semantically diverse. In practice, it is rare to encounter texts that are entirely outside of its scope. The table below with sampled topics of our five diverse datasets illustrates that a broad range of universal topics are captured:
>
> | Dataset | Sample topic 1 | Sample Topic 2 |
> |--|--|--|
> | PubMed | microscopy/… | rna/gene/… |
> | Arena | gameplay/rpg/…    | volcano/… |
> | Chat | comedian/…  | oscar/… |
> | Yelp | restaurant/…    | theater/… |
> | OpenReview | grammar/… | computational/… |
>
> The "out-of-domain" text is also considered and processed in our experiments--in our topic model, there is an “unclassified” bin for samples that are not close to any topic. The table below shows the details
>
> | Dataset | Training size | unclassified samples | Example |
> |-|:-:|:-:|--|
> | PubMed | 75,316 | 0 | |
> | Arena | 180,000 | 86 | i have no idea who am i, maybe you can help? |
> | Chat | 17,940 | 0 | |
> | Yelp | 1,939,290 | 4 | I am unable to provide check number. |
> | OpenReview | 8,396 | 0 | |
>
> We can see the number of unclassified samples is neglectable across all datasets, and those unclassified samples typically contain little to no substantive content, confirming the broad coverage of our topic model.
>
> ### Determine the number of clusters
> First, the number should not be too small, so as to keep the topic conditions meaningful. Second, it should not be too large, as that would amplify per-item DP-histogram noise in downstream tasks. To strike a balance, we set a threshold that each cluster must contain at least 100 out of 6M Wikipedia articles, resulting in 1,300 clusters.
>
>
> ### Quality guarantee of the proposed method
> Our framework introduces a universal topic model and controllability pretraining, which enables the model to capture valuable high-level, topic-wise information in the private domain—beyond the word-level representations learned by the vanilla DP-finetuning baseline. As a result, the data quality in our method should be guaranteed no worse than that of vanilla DP-finetuning. This quality improvement is empirically supported by the performance gains demonstrated in our experiments and further validated by our ablation study in Section 4.3.2.

---

> > ### Comment · Reviewer_qpMn · 2025-04-06
> >
> > Thanks for your response. I would like to keep my positive score. Good luck.

---

### Official Review · Reviewer_pWe8 · 2025-03-13

**Overall Recommendation:** 4

**Summary:**

This paper presents CTCL, a framework that synthesizes privacy-preserving data by combining a lightweight 140M parameter generator with a clustering-based topic model. The generator is differentially privately fine-tuned on private data, while the topic model produces a DP topic histogram to capture high-level distributional information. This approach circumvents the computational burden of fine-tuning billion-scale LLMs and the need for extensive prompt engineering, proving effective across diverse domains including sensitive applications and generative tasks.

**Claims And Evidence:**

Yes. The submission’s claims are largely supported by a comprehensive set of experiments and analyses. It demonstrates that CTCL outperforms several baselines on both generative and classification tasks under differential privacy constraints, and the scalability studies show clear improvements with larger synthetic datasets and higher privacy budgets.

**Essential References Not Discussed:**

NA

**Experimental Designs Or Analyses:**

NA

**Methods And Evaluation Criteria:**

Yes. The use of diverse benchmark datasets—ranging from academic medical texts to conversational dialogues and business reviews—ensures that the evaluation reflects real-world applications where privacy is critical. Moreover, evaluation metrics such as next-word prediction accuracy, perplexity, and MAUVE scores, along with thorough ablation studies and scalability experiments, provide robust evidence for the framework's effectiveness.

**Other Comments Or Suggestions:**

NA

**Other Strengths And Weaknesses:**

**Strengths:**

- The novel integration of a lightweight DP-finetuned generator with a clustering-based topic model enables efficient synthesis of high-quality synthetic data by capturing both fine-grained details and high-level distributional patterns.
- The approach leverages well-established DP techniques to provide robust privacy guarantees without requiring extensive prompt engineering or resource-intensive fine-tuning of large-scale models.



**Weakness:**
- My main concern is that paper lacks evaluation of the proposed approach against practical privacy attacks that might occur in real-world scenarios. So while the paper shows good utility improvements, its privacy aspects remain insufficiently evaluated.

**Questions For Authors:**

See weaknesses part. If the authors can show at least one type of attack (either a reconstruction attack or an attribute inversion attack), it would strengthen the paper.

**Relation To Broader Scientific Literature:**

CTCL innovatively integrates a lightweight, 140M-parameter conditional generator with a clustering-based topic model to capture both high-level distributional information and fine-grained textual details. This combination not only circumvents the resource constraints associated with billion-scale models but also offers enhanced controllability and scalability, marking a significant advancement in privacy-preserving data synthesis.

**Theoretical Claims:**

Yes. The algorithm is a chain of well-known DP algorithms. By the composition law of DP, everything is correct.

---

> ### Author Rebuttal · Authors · 2025-04-01
>
> Thank you for your supportive and detailed comments! We are encouraged by your feedback that our approach is novel, our experiments and analyses are comprehensive, and that our method makes  significant advancements in privacy-preserving data synthesis.
>
> ### Privacy Attacks
>
> According to the post-processing property of differential privacy (DP), our synthetic data has the same formal DP guarantees—$\epsilon=1,2,4$—as our generator, and these privacy budgets are widely accepted in the machine learning community, with $\epsilon < 10$ considered reasonable and $\epsilon \approx 1$ regarded as offering strong privacy protection ([Ponomareva et al. 2018](https://arxiv.org/abs/2303.00654)).
>
> DP already offers provable protection against membership inference and other, typically weaker, attacks such as reconstruction or attribute inversion attacks. This is supported by extensive empirical evidence, e.g., [Steinke et al. 2023](https://arxiv.org/abs/2305.08846), [Andrew et al. 2023](https://arxiv.org/abs/2302.03098), and [Nasr et al. 2023](https://arxiv.org/abs/2302.07956).
>
> Furthermore, recent works such as [Yue et al. 2023](https://arxiv.org/abs/2210.14348) and [Yu et al. 2024](https://arxiv.org/abs/2402.13659)’s experiments on secret sharer-style attacks ([Carlini et al. 2018](https://arxiv.org/abs/1802.08232)) in the DP synthetic data setting further support the privacy protection offered by DP.
>
> As practical attacks are usually weaker than the worst-case attacks that DP is designed to protect against, based on empirical privacy auditing and attack studies in recent years, we skipped auditing in this paper. That being said, we would be happy to include relavant experiments if you have a specific attack in mind that is beyond our consideration and feasible in our setting!

---

### Official Review · Reviewer_hXAS · 2025-03-14

**Overall Recommendation:** 4

**Summary:**

This paper introduces CTCL (Data Synthesis with Controllability and Clustering), a framework to generate privacy-preserving synthetic text data without fine-tuning LLMs or extensive domain-specific prompt engineering. The CTCL framework consists of two primary components: a lightweight 140M-parameter conditional text generator and a universal clustering-based topic model, both pretrained on publicly available datasets. To adapt to private domains, the generator undergoes differential privacy (DP) finetuning to capture fine-grained textual details, while the topic model constructs a DP-protected histogram to represent high-level distributional information. Synthetic data generation then leverages this DP histogram to control topic distribution during sampling. Empirical evaluations demonstrate that CTCL outperforms prior methods, such as Aug-PE, in benchmarks including PubMed and Chatbot Arena, particularly under tight privacy budget.

**Claims And Evidence:**

The claims made in the submission are generally well-supported by clear experiments and detailed analyses. The authors provide extensive experimental comparisons against several established baseline methods across multiple downstream tasks—both generative and classification—and demonstrate improvements, particularly under strict DP budget constraints.

**Essential References Not Discussed:**

N/A

**Experimental Designs Or Analyses:**

I reviewed the experiment design, especially the validation dataset. I like the inclusion of open-ended tasks in Chatbot Arena, which are usually not present in other LLM DP fine-tuning papers.

**Methods And Evaluation Criteria:**

The proposed methods make sense conceptually. The evaluation criteria follow prior works in DP finetuning and DP data synthesis.

**Other Comments Or Suggestions:**

Please see the questions below.

**Other Strengths And Weaknesses:**

**Strength**

The approach to synthesizing privacy-preserving data using a lightweight language model and clustering-based topic model is

**Weakness**
+ More design choices of the conditional generator are not explored.
+ The evaluation primarily focuses on quantitative benchmarks, with limited qualitative analysis of generated synthetic data. Additional qualitative analysis could help the reader further understand the practical usefulness and readability of the synthetic data.

**Questions For Authors:**

1. Since the proposed framework requires training a clustering-based topic model and a generator LLM on Wikipedia, how sensitive is it to domain mismatch between Wikipedia topics and the private datasets? Are there failure cases where this approach may struggle from domain mismatch?
2. Could the authors clarify how the DP budget was allocated between the DP Topic Histogram and DP Finetuning?
3. For the sample generation (4.1.3), the authors used nucleus sampling with top-p=0.95. How sensitive is the quality of generated data to the choice of sampling parameters?
4. What’s the rationale for using a 140M BART-like model? I think the paper can be stronger if more design choices for the generator are explored.

**Relation To Broader Scientific Literature:**

The contributions of this paper build specifically upon prior work in privacy-preserving data generation and differential privacy (DP) in LLMs. By integrating a lightweight (140M-parameter) conditional text generator with a universal topic model pretrained on public corpora, CTCL efficiently synthesizes high-quality text data under strict DP constraints without extensive prompt engineering or significant computational resources.

**Theoretical Claims:**

N/A

---

> ### Author Rebuttal · Authors · 2025-04-01
>
> Thank you for your encouraging remarks that our claims are well-supported by clear experiments and detailed analyses, and our proposed method is reasonable.
>
> ### Generator design choices
> We use a seq2seq generator because its encoder is well-suited for understanding the conditions. For size, the 140M model is to balance the efficiency with generation ability, making it practical for resource-constrained scenarios.
>
> We agree other design choices might perform better, but we do not extensively explore all designs because our intuitive choice already demonstrates strong downstream performanceas in our experiments. Moreover, same as all pretraining research, due to the cost and scale, it could be super challenging to exhaustively search over all possible configurations.
>
> We would also like to emphasize that the core contribution of our work is the overall CTCL framework. Each module within the framework remains flexible and open to further design improvements.
>
>
> ### Qualitative Analysis
> Thank you for the suggestion! We will include more qualitative examples in the next version of the paper in addition to those in Table 6. As noted in Section 4.3.3, we emphasize qualitative results because our primary usage of synthetic data is to support downstream model development. Therefore, the surface form of the synthetic text is less critical. For instance, Table 6 shows that PE-based generation using ChatGPT consistently produces fluent outputs. However, its downstream model performance is only comparable to (and sometimes worse than) that trained on less fluent synthetic data generated by DP-finetuned BART-base.
>
>
> ### Topic model design choices
> We choose Wikipedia for our topic model because it is large-scale, high-quality, and semantically diverse. In practice, it is rare to encounter texts that are entirely outside of its scope. The table below with sampled topics of our five diverse datasets illustrates that a broad range of universal topics are captured:
>
> | Dataset            | Sample topic 1        | Sample Topic 2         |
> |--------------------|-----------------------|------------------------|
> | PubMed             | microscopy/electron/… | rna/mir/gene/…         |
> | Chatbot Arena      | gameplay/rpg/wii/…    | eruptions/volcano/…    |
> | Multi-Session Chat | comedian/presenter/…  | oscar/nominations/…    |
> | Yelp               | restaurant/diner/…    | theater/cinemas/…      |
> | OpenReview         | grammar/syntax/…      | computational/turing/… |
>
> The "out-of-domain" text is also considered and processed in our experiments--in our topic model, there is an “unclassified” bin for samples that are not close to any topic. The table below shows the details
>
> | Dataset | Training size | unclassified samples | Example |
> |-|:-:|:-:|--|
> | PubMed | 75,316 | 0 | |
> | Chatbot Arena | 180,000 | 86 | i have no idea who am i, maybe you can help? |
> | Multi-Session Chat | 17,940 | 0 | |
> | Yelp | 1,939,290 | 4 | I am unable to provide check number. |
> | OpenReview | 8,396 | 0 | |
>
> We can see the number of unclassified samples is neglectable across all datasets, and those unclassified samples typically contain little to no substantive content, confirming the broad coverage of our topic model.
>
> ### Privacy Budget Allocation
> Our budget allocation follows [1], our "Post-Generation Resample" baseline, which also has both DP-histogram and DP-finetune steps. For DP-histogram, we adopt the same -- adding a Gaussian noise 10 on every histogram item. The overall privacy budget (ε) is computed as a composition of both steps using the standard `dp_accouting` package.
>
> We also show individual ε values below on PubMed, under composed ε=4, 2, or 1. DP-histogram only takes a small portion of the overall budget. This aligns with observations in [1] (page 5) and highlights our CTCL-topic design achieves strong performance while consuming only a small portion of the privacy budget.
>
> | ε (Composed) | ε (DP-Histogram) | ε (DP-Finetune) |
> |:-:|:-:|:-:|
> | 4 | 0.39 | 3.96 |
> | 2 | 0.39 | 1.94 |
> | 1 | 0.39 | 0.9 |
>
> We also find that the impact of changing the allocation is insignificant. For example, in ε=4 setting on PubMed, increasing histogram's noise from 10 to 20 only alters the DP-finetuning budget $ε$ marginally—from 3.96 to 3.99.
>
>
>
> ### Nucleus Sampling
> Our choice of nucleus sampling with top-p = 0.95 follows the standard practice introduced in [2], which has been shown to provide a strong balance between diversity and coherence, outperforming alternatives such as top-k or beam search. Also, since this standard hyperparameter setting already yields strong performance, we chose not to exhaustively tune this particular top-p value.
>
> [1] Privacy-Preserving Instructions for Aligning Large Language Models, ICML 2024
>
> [2] The Curious Case of Neural Text Degeneration, ICLR 2020.

---

> > ### Comment · Reviewer_hXAS · 2025-04-04
> >
> > I thank the authors for the detailed response. I will keep my score as this is already very positive, but I believe that the follow-up experiments further strengthen this paper.

---

### Official Review · Reviewer_E1ix · 2025-03-16

**Overall Recommendation:** 3

**Summary:**

The paper proposes a novel framework for generating privacy-preserving synthetic text data called CTCL. The authors claim that previous works mainly utilized fine-tuning the LLMs with differential privacy, which is computationally expensive or relies on extensive prompt engineering, which is time-consuming and does not necessarily result in a good performance.

Instead, CTCL includes several partitions to guarantee better performance and privacy at the same time. First, they use public data (here Wikipedia) to train a generator. Then using DP fine-tuning they adapt the generator to private data. The generator then synthesizes data guided by this DP histogram information of data. Evaluations across various domains demonstrate CTCL's effectiveness, especially under stringent privacy conditions, and its superior scalability compared to existing techniques.

**Claims And Evidence:**

YES

**Essential References Not Discussed:**

NO

**Experimental Designs Or Analyses:**

Yes

**Methods And Evaluation Criteria:**

YES

**Other Comments Or Suggestions:**

* There should be a cost comparison between the CTCL and the prior works. PE only used the model API to generate the synthetic data, but the CTCL pipeline requires training and fine-tuning multiple components.

* The authors have shown that changing the model API can change the performance of the PE method. The same analysis is applicable to the use of Gemma 2 in CTCL.

**Other Strengths And Weaknesses:**

STRENGTHS:

* The paper is very well-written. Motivation, prior works, and method intuitions are adequately discussed.

* The paper correctly identifies the shortcomings of the SOTA method (PE) and shows it in the experiments. (The most interesting result is that the PE's performance is primarily independent of the privacy budget).

* The final generator has only 140M parameters, which is efficient in fine-tuning and inference.

* The effort toward **reducing** the reliance on prompt engineering is appreciated.

WEAKNESS:

* There is an assumption that the public data can capture the primary information of the private text.

* The computational cost of the methods is missing. Some components seem very complex and expensive (although they are done once, it is good to understand the complexity and performance trade-off).

* The framework is complex; the authors have explained it in Figure 1, but a pseudo-code description can help the readers understand each step better.

**Questions For Authors:**

* The limitation section is missing.

* How is the method performed for out-of-domain private data?

**Relation To Broader Scientific Literature:**

* Improving the state-of-the-art DP-synthetic data generation. The synthetic data can later be used in the fine-tuning of models to preserve privacy and avoid the problems of directly using DP-finetuning.

* Instead of directly using prompt engineering to synthesize the data, they utilize the LLM to cluster the data. In this way, the performance is not directly dependent on the prompts or models.

**Theoretical Claims:**

Not Applicable

---

> ### Author Rebuttal · Authors · 2025-04-01
>
> We sincerely appreciate your recognition of the adequate discussion in our paper writing, the accurate identification of limitations in prior approaches, the efficiency of our lightweight final model, and our efforts to reduce reliance on prompt engineering.
>
> ### Public Data capturing private domain info.
> We agree a key intuition behind our design is the information learned from large-scale, diverse public data can benefit downstream learning in private domains. This aligns with the principle behind the pretraining–finetuning paradigm adopted across most LLM research.
>
> ### Topic model design choices
> We choose Wikipedia for our topic model because it is large-scale, high-quality, and semantically diverse. In practice, it is rare to encounter texts that are entirely outside of its scope. The table below with sampled topics of our five diverse datasets illustrates that a broad range of universal topics are captured:
>
> | Dataset | Sample topic 1 | Sample Topic 2 |
> |--|--|--|
> | PubMed | microscopy/… | rna/gene/… |
> | Arena | gameplay/rpg/…    | volcano/… |
> | Chat | comedian/…  | oscar/… |
> | Yelp | restaurant/…    | theater/… |
> | OpenReview | grammar/… | computational/… |
>
> The "out-of-domain" text is also considered and processed in our experiments--in our topic model, there is an “unclassified” bin for samples that are not close to any topic. The table below shows the details
>
> | Dataset | Training size | unclassified samples | Example |
> |-|:-:|:-:|--|
> | PubMed | 75,316 | 0 | |
> | Arena | 180,000 | 86 | i have no idea who am i, maybe you can help? |
> | Chat | 17,940 | 0 | |
> | Yelp | 1,939,290 | 4 | I am unable to provide check number. |
> | OpenReview | 8,396 | 0 | |
>
> We can see the number of unclassified samples is neglectable across all datasets, and those unclassified samples typically contain little to no substantive content, confirming the broad coverage of our topic model.
>
> ### Privacy Budget Allocation
> Our budget allocation follows [1], our "Post-Generation Resample" baseline, which also has both DP-histogram and DP-finetune steps. For DP-histogram, we do the same -- adding a Gaussian noise 10 on every histogram item. The overall privacy budget (ε) is computed as a composition of both steps using the standard `dp_accouting` package.
>
> We also show individual ε values below on PubMed, under composed $ε=4, 2, 1$. DP-histogram only takes a small portion of the overall budget. This aligns with observations in [1] (page 5) and highlights our CTCL-topic design achieves strong performance while consuming only a small portion of the privacy budget.
>
> | ε (Composed) | ε (DP-Histogram) | ε (DP-Finetune) |
> |:-:|:-:|:-:|
> | 4 | 0.39 | 3.96 |
> | 2 | 0.39 | 1.94 |
> | 1 | 0.39 | 0.9 |
>
> We also find that the impact of changing the allocation is insignificant. For example, in $ε=4$ setting on PubMed, increasing histogram's noise from 10 to 20 only alters the DP-finetuning budget $ε$ marginally—from 3.96 to 3.99.
>
>
> [1] Privacy-Preserving Instructions for Aligning Large Language Models, ICML 2024
>
> ### Computational Cost
> Our pretraining has 3 steps:
> 1. 2B LLM inference on a 430M documents.
> 2. Training a 140M LM on 430M (condition, document) pairs.
> 3. Generating embeddings for 6M Wikipedia docs with a 20M embedding model, followed by a HDBSCAN clustering.
>
> Among these, Step 1 dominates the computation due to the 2B LLM. This remains lighter than or comparable to the computation of pretraining a 2B LLM.
>
> Notebly, we will release our pretrained models, so users don't take the pretraining cost. This corresponds to the pretraining of the LLMs behind the APIs in PE-based approaches.
>
> For the downstream stage, PE methods often have significant user-side costs. For instance, AUG-PE synthesizes 2,000 samples in T=10 evolution iterations, each with L=4 variations, totaling 80K (2K x 10 x 4) ChatGPT requests. This represents a substantial API cost, whereas our approach applies finetuning on the lightweight model on local devices, which has no significant cost overhead.
>
> ### Use of Gemma
> As shown in our paper Table 1 and mentioned in Sec 3.1, unlike PE’s reliance on strongest LLMs like ChatGPT, our framework only requires basic instruction-following ability of the LLM, since aspects are extracted from existing documents without relying on LLM’s creativity.
>
> Notably, the LLM we use, Gemma-2-2B, is already one of the smallest LLMs. Larger models, such as its 9B and 27B variations, would likely yield even stronger results. We don’t experiment with more LLM choices because the small 2B LLM already has strong performance.
>
> We would also like to emphasize that the core contribution of our work is the overall CTCL framework. Each module design within the framework remains flexible and open to further design improvements.
>
> ### Pseudo-code and Limitations section
> Thank you for the suggestions! While ICML does not require a Limitations section, we will include both our pseudo-code and a limitation section into our next paper version for improved clarity.

---

### Official Review · Reviewer_QqU8 · 2025-03-16

**Overall Recommendation:** 3

**Summary:**

This paper introduces a new method CTCL to enable efficient synthetic data generation through privately fine-tuning a controlled generation LM. CTCL has two stages, first learning the general topics of private data through DP histogram learning and then privately fine-tunes a language model given the topics as the condition. CTCL only requires a LM of size 140M which is considerably smaller than existing on-device LLMs in the scale of billions.

**Claims And Evidence:**

The claims are supported by the evidence under the assumptions that the author made.

**Essential References Not Discussed:**

N/A

**Experimental Designs Or Analyses:**

The experiments are thorough to demonstrate the effectiveness of the proposed approach.

**Methods And Evaluation Criteria:**

The evaluation criteria makes sense to demonstrate the quality of synthetic data.

**Other Comments Or Suggestions:**

N/A

**Other Strengths And Weaknesses:**

Strengths:
1. The paper is well-written and motivated, privately generating high quality synthetic data with a small model is desirable especially considering fine-tuning on the private data on edge devices which have limited resources.
2. The experiments are thorough with multiple baseline approaches, tasks, and datasets considered.

Weaknesses:
1. It is unjustified why pre-training from scratch is needed for the LM part. Can one start from a pre-trained smaller language model with a similar number of parameters? Also, would knowledge distillation from an LLM into a smaller LM be enough?
2. There is an under-explained discrepancy between how condition is generated in the pretraining and fine-tuning stage. In the pre-training stage, the condition is extracted for each document from Gemma 2B, while for private data, the condition is from the topic model. Why not also use the topic model to generate the condition for the pretraining data as well? Or why not use Gemma 2B to extract aspects for the private data and learn those aspects privately?

**Questions For Authors:**

1. The generated conditions seem to be helpful for other methods as well, e.g. for private evolution, the conditions can be used for engineering the prompts for variations. Have the authors considered this as an alternative?
2. The topic model is trained on Wikipedia, how would it work when the topics from private data are personal and out of the domain of Wikipedia?

**Relation To Broader Scientific Literature:**

The paper is related to private synthetic data generation which is an important topic in the age of LLMs. The method targets resource-limited scenarios which match closely with real-world deployment.

**Theoretical Claims:**

N/A

---

> ### Author Rebuttal · Authors · 2025-04-01
>
> Thank you for your thoughtful comments and encouraging feedback that the problem we investigate is important, that our method is well-motivated and matching real-world applications, that our experiments are thorough, and that our paper is well-written!
>
> ### Pretraining
> We would like to clarify that our pretraining *does not* start from random initialization. As noted in Sec. 3.1, we perform *continual pretraining* on top of BART-base [1], a model already pretrained on a general corpus and it possesses basic language understanding abilities. Our continual pretraining is to endow the model with controllability.
>
> We agree that knowledge distillation could be an alternative to our pretraining, while it still falls within our overall framework, as it also needs properly constructed data in order to achieve controllability. Moreover, it should require more meticulous design choices and more computation due to the large teacher model.
>
> We would also like to emphasize that the core contribution of our work is the overall CTCL framework. Each module design within the framework remains flexible and open to further optimization and design improvements.
>
> [1] BART: Denoising Sequence-to-Sequence Pre-training for Natural Language Generation, Translation, and Comprehension, ACL 2020
>
>
> ### Discrepancy in constructed conditions between pretraining and finetuning
>
> **Why not topic model for pretraining?**
> The distinction between pretraining and finetuning conditions is intentional for two purposes: enhancing model generalizability and adhering to privacy constraints. In pretraining, using conditions from 2B LLM is for the diversity and flexibility of the aspects that the model can handle, enabling stronger controllability. Otherwise, “keywords” would become the only aspect in the pretraining.
>
> **why not Gemma 2B for private data?**
> If Gemma 2B were applied directly to private data, the extracted conditions would be in free-text format. This raises our concerns about sending the private information to the synthetic data generation process. In contrast, our framework ensures differential privacy by using a noised, topic-wise histogram instead. This preserves meaningful high-level topic information of the private domain while satisfying privacy requirements.
>
> To demonstrate the value of the flexible conditioning in pretraining, consider one may have a bit prior knowledge about the downstream domain, e.g., knowing the data is dialogues. In our framework, such prior knowledge can be incorporated, providing a better initialization for DP-finetuning.
>
> Below are the model generations controlled by doc type using the same keywords (*Surfing, World Championship, Young Athletes*), showcasing the better initialization:
>  * (Doc Type: News): *A member of WSA and WSL, won the World Championships for the first time in his career. [...]*
>  * (Doc Type: Chat): *How do you feel about the current state of surfing in the U.S. right now? \n There are a lot of great young athletes [...]*
>  * (Doc Type: Blog Post): *I've been fortunate enough to have the opportunity to meet the most talented young athletes [...]*
>
> This flexible design also intends to encourage broader and more creative uses of our model.
>
> ### Adapting condition generation to PE methods
> We agree our condition generation could be potentially adapted by PE, for example, within the variation prompts. However, as noted in the paper, a key limitation of PE methods is not learning fine-grained info from private data. Applying our condition generation into the PE framework might not address this key limitation of theirs.
>
> ### Topic Model on Wikipedia and Out-of-Domain Generalization
> We choose Wikipedia for our topic model because it is large-scale, high-quality, and semantically diverse. In practice, it is rare to encounter texts that are entirely outside of its scope. The table below with sampled topics of our five diverse datasets illustrates that a broad range of universal topics are captured:
>
> | Dataset | Sample topic 1 | Sample Topic 2 |
> |--|--|--|
> | PubMed | microscopy/… | rna/gene/… |
> | Arena | gameplay/rpg/…    | volcano/… |
> | Chat | comedian/…  | oscar/… |
> | Yelp | restaurant/…    | theater/… |
> | OpenReview | grammar/… | computational/… |
>
> The "out-of-domain" text is also considered and processed in our experiments--in our topic model, there is an “unclassified” bin for samples that are not close to any topic. The table below shows the details
>
> | Dataset | Training size | unclassified samples | Example |
> |-|:-:|:-:|--|
> | PubMed | 75,316 | 0 | |
> | Arena | 180,000 | 86 | i have no idea who am i, maybe you can help? |
> | Chat | 17,940 | 0 | |
> | Yelp | 1,939,290 | 4 | I am unable to provide check number. |
> | OpenReview | 8,396 | 0 | |
>
> We can see the number of unclassified samples is neglectable across all datasets, and those unclassified samples typically contain little to no substantive content, confirming the broad coverage of our topic model.

---

### Decision · Program_Chairs · 2025-05-01

**Decision:**

Accept (poster)

**Comment:**

The biggest strength of the paper is that it performs better than prior work, as presented in Table 3. However, the main concern is that the method closely resembles technique proposed in [1], where the generator is DP-finetuned on private data and the DP-histogram is used to better align the distribution of the generated samples with the private data. Both methods follow this two stage approach. All Reviewers lean towards accept.